# Emergent Hierarchical Reasoning in LLMs through Reinforcement Learning

**Haozhe Wang**♣♠∗**, Qixin Xu**◇**, Che Liu**♡**, Junhong Wu**ϑ**, †Fangzhen Lin**♣**, †Wenhu Chen**♠

♣Hong Kong Univerisity of Science and Technology, ♠University of Waterloo
∗M-A-P, ◇Tsinghua Univerisity, ♡Imperial College London, ϑUCAS
{jasper.whz@outlook.com, wenhuchen@uwaterloo.ca}
⌂ https://tiger-ai-lab.github.io/Hierarchical-Reasoner/

## Abstract

Reinforcement Learning (RL) has proven highly effective at enhancing the complex reasoning abilities of Large Language Models (LLMs), yet underlying mechanisms driving this success remain largely opaque. Our analysis reveals that puzzling phenomena like "aha moments", "length-scaling" and entropy dynamics are not disparate occurrences but hallmarks of an emergent reasoning hierarchy, akin to the separation of high-level strategic planning from low-level procedural execution in human cognition. We uncover a dynamic evolution where the learning bottleneck shifts: initially, the process is dominated by procedural consolidation and must improve its low-level skills. The learning bottleneck then decisively shifts, with performance gains being driven by the exploration and mastery of high-level strategic planning. This insight exposes a core inefficiency in prevailing RL algorithms like GRPO, which apply optimization pressure agnostically and dilute the learning signal across all tokens. To address this, we propose Hierarchy-Aware Credit Assignment (HICRA), an algorithm that concentrates optimization efforts on high-impact planning tokens. Our extensive experiments validate that HICRA significantly outperforms strong baselines, and offer deep insights into how reasoning advances through the lens of strategic exploration.

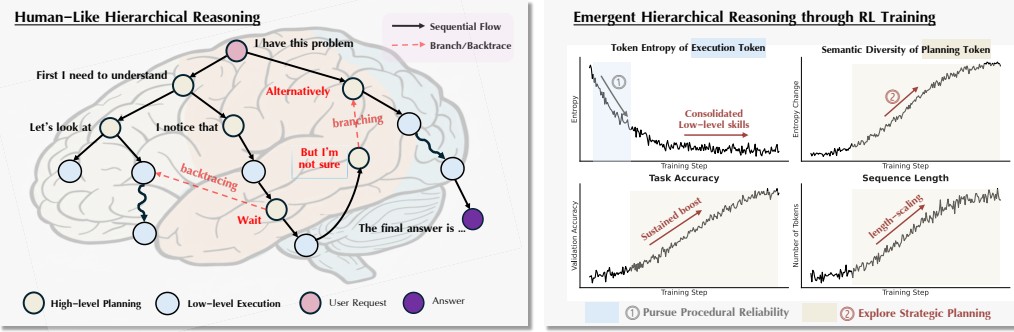

Figure 1: **(Left)** LLM reasoning mirrors a human-like hierarchical reasoning: high-level strategic planning and low-level procedural executions. **(Right)** Hierarchical reasoning emerges during RL training via a two-phase dynamic. Phase ① consolidates low-level skills, marked by a token-entropy drop in execution tokens. The learning frontier then shifts to Phase ②, where the model explores and masters high-level planning, marked by increased semantic diversity, sustained reasoning enhancement and length scaling.

## 1 Introduction

Reinforcement Learning (RL) has become instrumental in advancing the complex reasoning capabilities of Large Language Models (LLMs) across diverse domains (Ouyang et al., 2022; Jaech et al., 2024; Yang et al., 2024; Guo et al., 2025; Team et al., 2025). However, this empirical success is accompanied by a significant gap in our understanding of the underlying learning dynamics. The training process often yields phenomena that are as effective as they are poorly understood: models

can experience sudden 'aha moments', where they seemingly acquire new emergent skills (Guo et al., 2025); they exhibit 'length-scaling' effects, where reasoning performance improves with longer, more detailed outputs (Guo et al., 2025; Team et al., 2025); and they display complex dynamics in token-level entropy (Yu et al., 2025; Cui et al., 2025). This gap motivates a fundamental question:

**What unlocks enhanced reasoning in LLMs during RL, and how should we leverage this understanding to design more principled and efficient RL algorithms?**

Our investigation is guided by a key insight: RL does not train models de novo. It fine-tunes base models already imbued with **priors** from pre-training on vast corpora of human-written solutions. These solutions inherently encodes the **hierarchical structure of human reasoning** – a highly efficient cognitive strategy evolved under biological constraints. This prompts us to ask: does RL unlock advanced reasoning by (re-)discovering this hierarchical structure as a promising pathway for solving math problems?

To test this hypothesis, we analyze the RL training process through the lens of hierarchical reasoning. Drawing a parallel to the cognitive architecture of the human brain (Fig. 1), which separates high-level, deliberate strategic planning from the rapid execution of learned procedures (Murray et al., 2014; Zeraati et al., 2023; Huntenburg et al., 2018), we propose a decomposition of model-generated tokens into two functional hierarchy :

- **High-level Planning Tokens:** The high-level strategic moves that orchestrate the reasoning process. These tokens manifest as logical maneuvers, including deduction (e.g., "we can use the fact that"), branching (e.g., "let's try a different approach"), and backtracing (e.g., "but the problem mentions that").
- **Low-level Execution Tokens:** The operational building blocks of a solution. These comprise concrete, low-level steps such as arithmetic calculations, variable substitutions, and the direct application of known formulas.

Our analysis across eight text-only and vision-language models confirms this hypothesis, revealing a consistently two-phase dynamic that explains the **emergence** of this reasoning hierarchy in LLMs. We find the optimization pressure of RL is not static; instead, its learning frontier shifts. Initially, the process is constrained by **procedural correctness**. A single calculation error can invalidate an entire solution, creating a powerful learning signal that compels the model to first master low-level execution tokens. Once proficiency in these foundational skills is achieved, the learning bottleneck shifts to **strategic planning**. We find that these phases are not mutually exclusive; procedural refinement continues throughout training, but the primary driver of marginal performance gains shifts to strategic planning – exploring and mastering the use of planning tokens is what unlocks significant and sustained improvements in reasoning ability.

This emergent two-phase mechanism provides a unifying framework for the puzzling phenomena observed in RL training. It explains "aha moments" as the discovery and internalization of high-level strategic reasoning strategies, such as self-reflections. It also accounts for the "length-scaling" effect, as employing more sophisticated strategies – involving thorough planning and logical backtracing – naturally elongates the reasoning trace with structured, strategic deliberation. Notably, it provides a unified perspective to understand the complex token entropy dynamics across different models, through the lens of high-impact planning tokens and gradually confident execution tokens.

This discovery – that the learning frontiers dynamically shifts to strategic planning – is more than an academic curiosity; it provides a clear blueprint for a more effective RL algorithm. If the primary driver for advanced reasoning is the mastery of high-level strategic planning, then current agnostic credit assignment methods used in the prevailing GRPO (Guo et al., 2025) and its variants (Yu et al., 2025; Liu et al., 2025b; Wang et al., 2025c) are fundamentally inefficient, as they dilute optimization pressure across all tokens rather than concentrating it where it matters most.

Based on this insight, we propose **Hierarchy-aware Credit Assignment (HICRA)**, a novel algorithm designed to focus optimization pressure directly on this emergent strategic bottleneck. By selectively amplifying the learning signal for planning tokens, HICRA accelerates the exploration and reinforcement of effective high-level reasoning, leading to significant performance gains as demonstrated in our experiments.

**Contributions.** In this work, we advance the understanding of how LLMs learn to reason via RL. We demonstrate that the learning process is not monolithic but an emergent two-phase learning dynamic

driven by the hierarchical priors in base models and solution structure of the reasoning tasks. This insight reveals that the true bottleneck for advanced reasoning is the mastery of high-level strategic planning, which current agnostic credit assignment methods neglect. To bridge this gap, we pioneer with an original and simple solution, HICRA. Through extensive experiments across LLMs and VLMs, we not only validate the effectiveness of HICRA, but also offer deep insights into how HICRA works through the lens of strategic exploration.

## 2 THE EMERGENT REASONING HIERARCHY

**Guiding Insight:** The pre-training priors and the inherent structure of reasoning tasks create a strong inductive bias. For the task of math problem-solving, hierarchical reasoning proves an efficient and prominent strategy, which is discovered through RL training and unlocks advanced reasoning.

### 2.1 A FUNCTIONAL PROXY FOR THE REASONING HIERARCHY

To analyze this reasoning hierarchy, we must first distinguish high-level strategic planning from low-level procedural execution within the model's generated tokens. This is challenging because a token's function is defined by its context, not its intrinsic meaning.

To address this gap, we draw inspiration from human cognition. When a person reasons through a problem, we easily identify their strategic thinking by its function. A phrase like, "Let's try a different approach," functions as a high-level strategic maneuver that guides the problem-solving direction. In contrast, a phrase like, "so we add 5 to both sides," is a low-level procedural step. Inspired by this functional distinction, we introduce Strategic Grams as a functional proxy to circumvent the difficulty of formally defining what is a "planning token".

**Strategic Grams (SGs)** are defined as $n$-grams that function as a single semantic unit to guide the logical flow. We use n-grams because they capture the phrasal nature of strategic language (e.g., "let's consider the case") which is lost at the single-token level. These SGs facilitate three main types of logical moves: (a) deduction, (b) branching, and (c) backtracing, as we show in an illustrative example in the appendix.

*A token is classified as a strategic **planning token** if it is part of a Strategic Gram in the current context. All other tokens are classified as procedural **execution tokens**.*

For simplicity, we use the term "execution tokens" to refer to all non-planning tokens. We note this is a slight simplification, as this category encompasses not only concrete calculations but also formatting and other procedural language.

A key challenge is identifying the set of SGs in a principled and reproducible manner. Manual annotation or reliance on proprietary models would introduce subjectivity and hinder reproducibility. We therefore propose an automated, data-driven pipeline based on a key insight: SGs function as the reusable scaffolding of a reasoning process (Fig. 6). This function imparts a distinct statistical signature: SGs should appear frequently across a wide range of different solutions but be used sparingly within any single solution. However, a significant challenge is the linguistic diversity of strategic language, where a single strategic intent can be expressed through numerous phrases.

Our pipeline is designed to overcome these challenges by first grouping semantically equivalent n-grams and then identifying which consolidated concepts exhibit the statistical signature of strategic planning. We place the detailed construction procedure to the appendix due to page limits.

This automated procedure is designed to yield a high-precision functional proxy for strategic planning, not an exhaustive lexicon of all possible SGs. We set reasonable hyper-parameters for identifying SGs, and we contend that the resulting SG collection is sufficiently representative to reveal the core learning dynamics. To validate this claim, we conduct a sensitivity analysis by randomly removing 30% of the identified SGs and re-running our main analysis (see Appendix). The resulting learning dynamic curves remain qualitatively identical, demonstrating the robustness of our methodology and the findings derived from it. To validate that our automated pipeline captures genuine semantic intent rather than statistical noise, we conducted a human annotation study. Results confirm that 86% of our identified SGs were classified by humans as functioning to "guide flow or propose plans," compared to only 12% for otherwise. (See Appendix for full study details).

## 2.2 Emergence of the Reasoning Hierarchy

Building on our functional proxy for reasoning, we examine the learning dynamics of RL for LLM reasoning and finds an intriguing parallel with human-like hierarchical reasoning. Our empirical analysis – conducted consistently across different model families, Qwen2.5-7B (Yang et al., 2024), Qwen3-4B (Yang et al., 2025), Llama-3.1-8B (Grattafiori et al., 2024), Qwen2.5-VL-7B (Bai et al., 2025), MiMO-VL-7B (Xiaomi, 2025) – reveals that **enhanced reasoning is not a monolithic process, but driven by an evolution of the learning frontiers**.

The learning process exhibit two overlapping phases: it often begins with a rapid consolidation of procedural reliability, conducive to the widespread low-level tokens. This is followed by a sustained period where the greatest potential for improvement shifts to the exploration of high-level strategic reasoning, which serves as the true engine of advanced performance.

### 2.2.1 Forging Reliable Low-level Skills

The initial phase of RL training is dedicated to mastering the basics. The model must first build a reliable engine for low-level skills, e.g., formatting, performing calculations and other procedural steps. To observe this, we track two key metrics on the execution tokens:

- **Relative Perplexity:** Perplexity, the exponentiated average negative log-likelihood, measures model surprise. A lower value signifies higher confidence. We normalize the perplexity by its initial value to compare the rates of change in planning tokens and execution tokens.
- **Token-Level Entropy:** The Shannon entropy of the policy's next-token distribution, $H(\pi(\cdot|x_{<t}))$, measures its uncertainty. High entropy signals active exploration over the vocabulary at the next-token, while low entropy suggests confident exploitation.

The evidence for this phase is shown in the first two columns of Figure 2, marked with ①. The Relative Perplexity of execution tokens (grey curves) plummets in the early stages of training before flattening (column 1). This shows the model rapidly becomes confidently correct in its procedural steps. This is reinforced by the Token Entropy graph (column 2), where entropy for execution tokens is consistently and significantly lower than for planning tokens. The model is not just confident; it actively reduces exploration of procedural alternatives to converge on reliable operations. This rapid mastery of the basics is the first learning frontier to be solved.

> **Takeaway 1.** Procedural consolidation is often marked by a sharp decrease in the perplexity and token entropy of execution tokens. The model quickly builds a reliable "toolbox" of procedural skills, allowing the primary frontier for performance improvement to shift to high-level strategy.

Notably, we find that this phase of low-level skill consolidation might be absent or shot in models with stronger capacity, as evident in MiMO-VL-Instruct and Qwen-4B-Instruct. This also supports the argument that the primary driver of RL is indeed the exploration of strategic planning. We refer the reader to check the full analysis of training dynamics across eight models in the appendix.

### 2.2.2 Steering the Skills with Strategic Planning

Once the model becomes procedurally reliable, its performance gains are primarily driven by its ability to explore and deploy a diverse set of high-level strategies. To track this shift, we analyze the planning tokens using two key metrics. We compute the **Semantic Entropy** of strategic grams – the Shannon Entropy of the frequency distribution of strategic grams – to quantify the diversity of the model's high-level strategic plans (illustrated in Fig. 10). To isolate procedural variety, we compute the **conditional entropy** of subsequent procedural n-grams given a preceding strategic gram. This second metric shows how varied is the subsequent procedural steps for a preceding strategic move.

The third column of Figure 2 provides clear evidence of this strategic exploration phase. The semantic entropy of strategic grams (red line, marked with ②) shows a distinct and steady increase. This indicates that the model is not converging on a single optimal strategy but is instead actively expanding its repertoire of strategic plans. This observation is critical: mastery in reasoning, in this context, is achieved by developing a rich and varied strategic playbook, which contrasts sharply with the sharp decrease in token-level entropy seen during the initial procedural consolidation phase.

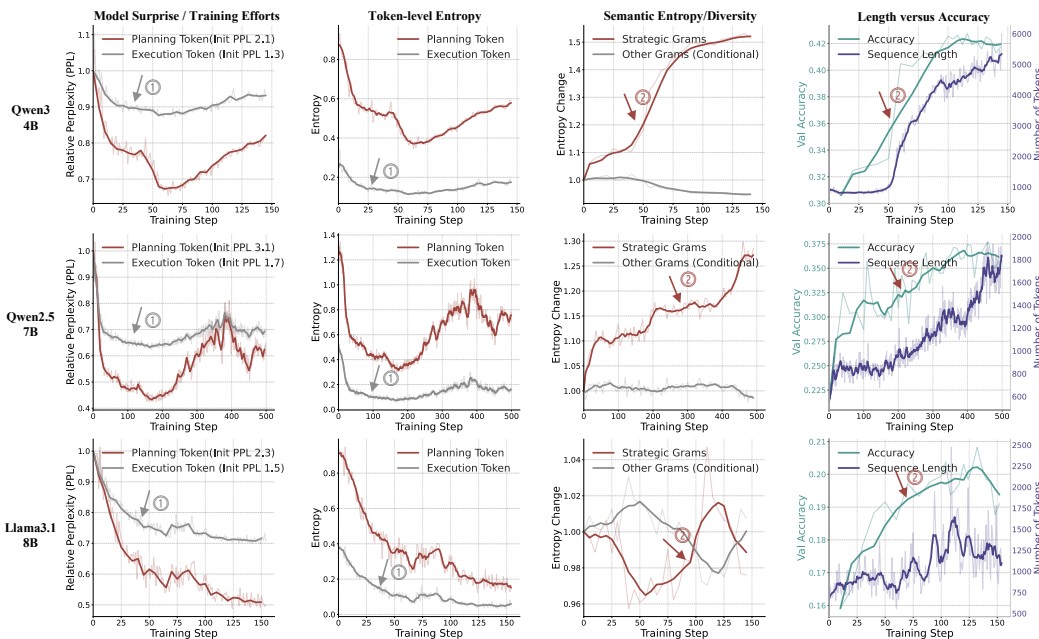

Figure 2: We track the training Dynamics of representative model families. The curves reveal a two-phase dynamics. Seen from the first two columns, the model has an initial focus on procedural consolidation, marked by sharp decrease in model perplexity (greater confidence) and token entropy (more certain) of execution tokens. This follows a shift to exploring strategic planning, evident from the third column. The diversity of strategic plans (semantic entropy) steadily increases on Qwen models or takes a turn to increase on Llama, correlating with consistently improved accuracy and longer reasoning chains (fourth column).

This strategic diversification provides the most direct evidence for our thesis: the model isn't just getting better at executing plans; it's getting better at planning itself. While the model explores new high-level strategic moves, the conditional entropy of procedural grams (grey line) remains stable. This suggests that once a procedural skill like arithmetic is mastered, there is little incentive to find diverse ways to perform it. The improved reasoning performance comes from discovering new ways to combine these established skills, which is the core function of strategic planning.

Crucially, *this expansion of the strategic playbook directly correlates with tangible performance gains*. The fourth column shows that the rise in strategic diversity is accompanied by a parallel increase in the length of reasoning chains and a sustained boost in overall accuracy. This demonstrates that after procedural skills are consolidated, the development of strategic planning becomes the primary bottleneck and driver for advanced reasoning performance.

> **Takeaway 2.** Once payoff from procedural consolidation diminishes, performance gains are driven primarily by exploring high-level strategies. This is marked by the increasing semantic diversity of strategic grams, which correlate with sustained reasoning enhancement, length scaling, and represents the key learning frontier.

**Explaining Puzzling Phenomena.** This emergent reasoning hierarchy provides a unified explanation for previously observed behaviors.

- **"Aha moments"** are the behavioral signature of the model discovering, mastering, and reinforcing a new, powerful strategy or set of strategic constructs.
- **"Length-scaling"** is highly consistent with increase in strategic diversity. As Figure 2 shows, the rise in semantic entropy of planning tokens is strongly correlated with an increase in average sequence length. More sophisticated strategies – involving planning, case analysis, and self-reflections – are mediated by planning tokens and naturally produce longer, more successful reasoning traces.

**Semantic Entropy: A Good Compass for Exploration.** The trends in Figure 2 also highlight a critical flaw in using aggregate token-level entropy (column 2) to track exploration.

> **Takeaway 3.** The aggregate token-level entropy is dominated by the vast majority of low-level execution tokens. As the model become confident in procedural steps or low-level skills, the entropy of these tokens naturally decreases, pulling the global average down.

Unluckily, the decrease in token-level entropy sometimes mislead practitioners into the conception of declined exploration. This is incorrect, however, as it contradicts the fact of increasing exploration in strategic plans (semantic entropy of planning tokens) and the improving reasoning performance. In the appendix, we compare with token entropy and Pass@K. Our results show that semantic entropy avoids the flaws in token entropy by directly measuring diversity at the semantic level of meaningful strategic units. Its trend accurately reflects the expansion of the model's strategic playbook, making it a more reliable diagnostic tool for tracking genuine exploration and predicting sustained performance improvements. It also complements Pass@K metric with further benefits. We refer interested readers to the appendix for a full analysis of training dynamics across eight LLM and VLMs and the deeper insights into RL training and exploration.

## 3 HICRA: HIERARCHY-AWARE CREDIT ASSIGNMENT

Our empirical analysis reveals a fundamental insight: RL improves reasoning by rediscovering and operationalizing the strategic layer of reasoning inherited from the model's pre-training priors. The learning process is characterized by a dynamic shift in its learning frontiers. Initially, the model is constrained by procedural correctness, but as it masters these foundational skills, the frontier for performance improvement shifts to the exploration and mastery of high-level strategic planning.

This observation exposes a core inefficiency in prevailing RL algorithms like GRPO, which apply optimization pressure agnostically across all tokens. Such methods fail to concentrate learning where it matters most – on the emergent strategic bottleneck. To address this, we propose an algorithm designed to focus the model's learning capacity on the sparse, high-impact planning tokens that orchestrate a successful reasoning trace.

**Formulation.** We introduce **Hierarchy-Aware Credit Assignment (HICRA)**, an algorithm that builds upon the GRPO framework to allocate credit based on the reasoning hierarchy. In GRPO, given a query $\mathbf{q}$ from a dataset $\mathcal{D}$, the policy $\pi_\theta$ generates a set of $G$ output trajectories $\{\mathbf{o}_1, \ldots, \mathbf{o}_G\}$. The advantage for a token $o_{i,t}$ at timestep $t$ in trajectory $\mathbf{o}_i$ is the group-normalized reward:

$$\hat{A}_{i,t} = R(\mathbf{q}, \mathbf{o}_i) - \frac{1}{G} \sum_{j=1}^{G} R(\mathbf{q}, \mathbf{o}_j)$$

HICRA, pronounced "high-krah", modifies this advantage to prioritize planning tokens. Let $\mathcal{S}_i$ be the set of indices corresponding to planning tokens within trajectory $\mathbf{o}_i$, identified using the method in Section 2.1. We define the HICRA advantage as:

$$\hat{A}_{i,t}^{\text{HICRA}} = \begin{cases} \hat{A}_{i,t} + \alpha \cdot |\hat{A}_{i,t}| & \text{if } t \in \mathcal{S}_i \\ \hat{A}_{i,t} & \text{if } t \notin \mathcal{S}_i \end{cases}$$

where $\alpha \in (0, 1)$ is a hyperparameter controlling the amplification intensity (we use $\alpha = 0.2$ in our experiments). This formulation creates a clear learning hierarchy: for successful trajectories ($\hat{A}_{i,t} > 0$), it amplifies the credits for planning tokens, while for unsuccessful ones ($\hat{A}_{i,t} < 0$), it dampens their penalty. The resulting RL objective and its policy gradient (simplified without PPO clipping) are:

$$\mathcal{J}(\theta) = \mathbb{E}_{\mathbf{q}\sim\mathcal{D}, \mathbf{o}_i\sim\pi_\theta} \left[ \hat{A}_{i,t}^{\text{HICRA}} \right], \quad \nabla\mathcal{J}(\theta) = \mathbb{E} \left[ \hat{A}_{i,t}^{\text{HICRA}} \cdot \nabla \log \pi_\theta(o_{i,t}|\mathbf{q}, \mathbf{o}_{i,<t}) \right]$$

By translating the amplified advantage into a stronger policy gradient, HICRA directly focuses the model's optimization on the strategic elements of its reasoning process. Unlike methods that reward statistical uncertainty (entropy) indiscriminately, HICRA targets the semantic function of planning. As we show in Section 4 and the Appendix, rewarding high entropy is systematically different from rewarding high-level planning.

**Connection to Strategic Exploration.** The core mechanism of HICRA engineers more effective exploration by reshaping the policy update's target distribution. A standard policy gradient (Williams,

2004) update nudges the policy $\pi_{\theta_{old}}$ toward an implicit target distribution $\pi^*$ defined by the advantage function described as follows (the derivation is included in the appendix):

$$\pi^*(o_{i,t}|\mathbf{q}, \mathbf{o}_{i,<t}) \propto \pi_{\theta_{old}}(o_{i,t}|\mathbf{q}, \mathbf{o}_{i,<t}) \exp(\hat{A}_{i,t})$$

Typically, this update pressure is applied isotropically, affecting all token types uniformly. HICRA breaks this symmetry. By using the modified advantage $\hat{A}^{\text{HICRA}}$, it creates a new target distribution, $\pi^*_{\text{HICRA}}$, that is anisotropically stretched toward the strategic dimensions of the action space. This new target distribution places significantly greater probability mass on planning tokens (through the term $\exp(\hat{A}_{i,t})$), particularly those within high-reward trajectories.

This anisotropic reshaping fosters a potent virtuous feedback loop: (a) the policy is incentivized to explore the subspace of strategic plans more thoroughly; (b) this leads to the faster discovery of effective reasoning patterns; and (c) when these strategies yield high rewards, the amplified advantage ensures they are strongly reinforced, cementing the model's planning capabilities far more efficiently. We also validate the effects of HICRA in exploration through experiments in Section 4.

## 4 EXPERIMENTS

**Models and Datasets.** Our experiments use open-source models including Qwen2.5-7B (Yang et al., 2024), Qwen3-4B (Yang et al., 2025), LLama-3.1-8B (Grattafiori et al., 2024), and VLMs like Qwen2.5-VL-7b (Yang et al., 2024) and MiMO-VL-7B (Xiaomi, 2025), covering both base and instruction-tuned variants. We train on established reasoning datasets DAPO (Yu et al., 2025), DeepScaleR (Luo et al., 2025) and ViRL39K (Wang et al., 2025c) for VLMs.

**Benchmarks and Baselines.** We evaluate on a suite of challenging text-only (e.g., AIME24, AIME25 (Mathematical Association of America, 2024), Math500 (Lightman et al., 2023), AMC23, Minerva (Lewkowycz et al., 2022), and Olympiad (He et al., 2024)) and multimodal (e.g., Math-Vista (Lu et al., 2023), MathVerse (Zhang et al., 2024), MathVision (Wang et al., 2024), EMMA (Hao et al., 2025)) benchmarks. We adopt the evaluation protocols of Deepseek R1, using Pass@1 with random samplings. We compare HICRA against three primary baselines: the **Base** model (before RL), the widely adopted **GRPO** baseline with clip-higher (Yu et al., 2025) by default, **Entropy Regularization**: GRPO with an additional regularization loss on token-level entropy (Cheng et al., 2025), **High-Entropy Advantage**: GRPO with advantage modulation on high-entropy tokens, following Cheng et al. and Wang et al., and **Placebo HICRA**: rewarding random n-grams.. A comprehensive description of our evaluation protocol, training implementation, and additional model-specific details can be found in the Appendix.

### 4.1 MAIN RESULTS

Our primary results, summarized in Table 1 and Table 3 in the appendix, show that **HICRA consistently and outperforms both the GRPO baselines** across text-only models and vision-language models on various benchmarks. On the strongest base model, Qwen3-4B-Instruct, HICRA's gains demonstrate that even on highly capable models, selectively amplifying the learning signal for strategic reasoning yields substantial improvements. This trend holds for non-instruct-tuned models as well, providing strong empirical evidence for our central claim: by identifying and focusing on the emergent strategic bottleneck, HICRA accelerates the development of advanced reasoning abilities more efficiently than agnostic methods.

### 4.2 ANALYSIS OF RL'S IMPACT ON REASONING

We conduct a series of analyses to dissect how RL improves reasoning. First, we have linked strategic planning to reasoning through analyses of the training dynamics in Section 2.2; Second, we verify the key effects of RL by showing the frequency dynamics of different errors throughout training (Section 4.2.1); we then justify the effectiveness of HICRA in exploration by comparing with standard entropy-regularized baselines. Further insights and analyses are included in the appendix.

Table 1: Comparison of HICRA, GRPO, and Base models across various mathematical reasoning benchmarks. HICRA consistently outperforms all baselines across different base models, demonstrating the effectiveness of focusing optimization on strategic planning tokens.

| Model | AIME24 | AIME25 | Math500 | AMC23 | Minerva | Olympiad |
|---|---|---|---|---|---|---|
| **Qwen3-4B-Instruct-2507** | | | | | | |
| Base | 63.4 | 47.7 | 94.6 | 86.7 | 45.2 | 72.4 |
| GRPO | 68.5 | 60.0 | 96.2 | 88.5 | 50.0 | 72.7 |
| HICRA | **73.1** | **65.1** | **97.2** | **90.2** | **50.7** | 72.0 |
| Δ (HICRA - GRPO) | +5.4 | +5.1 | +1.0 | +1.7 | +0.7 | -0.7 |
| **Qwen3-4B-Adaptive** (No-Think) | | | | | | |
| Base | 21.3 | 18.1 | 84.4 | 60.5 | 40.4 | 49.9 |
| GRPO | 63.1 | 58.8 | 95.6 | 76.8 | 45.2 | 55.6 |
| HICRA | **65.9** | **62.1** | **95.8** | **82.5** | **46.3** | **59.7** |
| Δ (HICRA - GRPO) | +2.8 | +3.3 | +0.2 | +5.7 | +1.1 | +4.1 |
| **Qwen3-4B-Base** | | | | | | |
| Base | 9.4 | 5.3 | 63.8 | 38.9 | 28.3 | 30.7 |
| GRPO | 24.9 | 23.8 | 83.0 | 51.2 | 38.9 | 45.8 |
| HICRA | **31.0** | **27.6** | **89.0** | **54.0** | **42.5** | **48.1** |
| Δ (HICRA - GRPO) | +6.1 | +3.8 | +6.0 | +2.8 | +3.6 | +2.3 |
| **Llama-3.1-8B-Instruct** | | | | | | |
| Base | 4.2 | 0.6 | 50.2 | 17.1 | 20.9 | 13.7 |
| GRPO | 8.9 | 0.5 | 53.0 | 25.0 | 27.2 | 20.3 |
| HICRA | 8.3 | **0.8** | **54.8** | **27.1** | 25.8 | **21.2** |
| Δ (HICRA - GRPO) | -0.6 | +0.3 | +1.8 | +2.1 | -1.4 | +0.9 |
| **Qwen2.5-7B-Base** | | | | | | |
| Base | 3.5 | 1.7 | 55.6 | 46.9 | 30.9 | 25.9 |
| GRPO (Guo et al., 2025) | 16.3 | 11.4 | 77.6 | 46.7 | 36.8 | 41.9 |
| ORZ (Hu et al., 2025) | 17.0 | 13.1 | 80.6 | 52.4 | 39.7 | 44.9 |
| SimpleRL (Zeng et al., 2025) | 16.7 | 3.3 | 78.2 | 42.8 | 34.9 | 38.2 |
| High-Entropy Advantage | 15.8 | 11.4 | 78.9 | 51.8 | 33.8 | 40.8 |
| Placebo HICRA | 14.6 | 9.3 | 77.4 | 51.8 | 34.8 | 41.2 |
| Entropy Regularization | 16.0 | 9.3 | 77.4 | 50.3 | 33.1 | 40.6 |
| HICRA | **18.8** | **14.8** | **80.2** | **55.1** | **38.6** | **45.9** |
| Δ (HICRA - GRPO) | +2.5 | +3.4 | +2.6 | +8.4 | +1.8 | +4.0 |

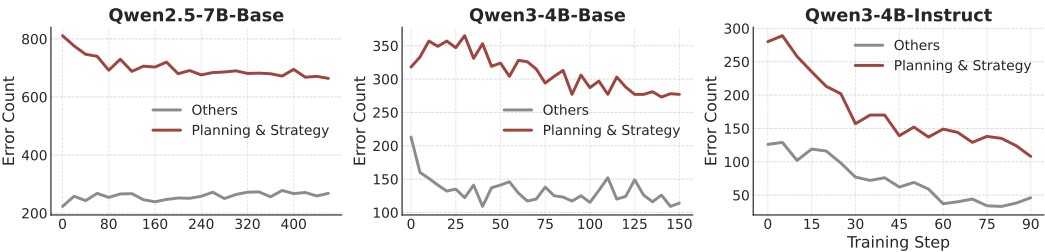

Figure 3: **Training Dynamics of Error Types.** Across all models, the number of *Planning & Strategy* errors (red) decreases more significantly than other procedural errors (gray), indicating that RL's primary benefit comes from correcting high-level strategic faults.

### 4.2.1 MASTERY OF STRATEGIC PLANNING UNLOCKS IMPROVED REASONING DURING RL

To understand where RL applies the most leverage, we analyzed the evolution of error types in failed rollouts. We first manually reviewed failures and nominated four distinct error causes. GPT-4o was then prompted to classify each failure into one of these causes via a multiple-choice question. Finally, we parsed these classifications into two broader categories: "Planning & Strategy" (e.g., flawed logic, incorrect high-level plan) and "Others" (e.g., calculation mistakes, fact-retrieval errors). The prompt used is included in the appendix.

Figure 3 reveals a consistent pattern: the primary benefit of RL stems from fixing high-level strategic faults. Across all models, the reduction in strategic errors is more pronounced than the reduction in other errors. This pattern is especially illuminating for Qwen2.5-7B-Base, where non-planning errors does not decrease. We conjecture that while the model may be improving its procedural reliability, these low-level enhancements do not translate to correct answers because the high-level strategy remains the limiting factor. A perfectly executed incorrect plan will still result in failure.

This evidence strongly supports our claim that **the strategic bottleneck is the key to unlocking advanced reasoning**. RL preferentially corrects these high-level faults over low-level execution mistakes, as improving strategic planning provides the most direct path to solving complex problems.

### 4.2.2 JUSTIFYING HICRA: TARGETED VS. INDISCRIMINATE EXPLORATION

Our findings suggest that performance gains are driven by mastering high-level strategic planning, which motivates HICRA's design to concentrate learning on planning tokens. As shown in Figure 4, HICRA's success is linked to its ability to sustain a higher level of *semantic entropy* than GRPO. This heightened diversity in high-level strategies directly correlates with stronger and more stable validation accuracy, confirming that focused strategic exploration is a primary driver of reasoning improvements.

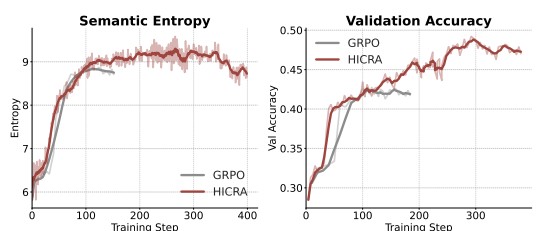

Figure 4: HICRA improves GRPO Clip-Higher via more diverse strategic exploration.

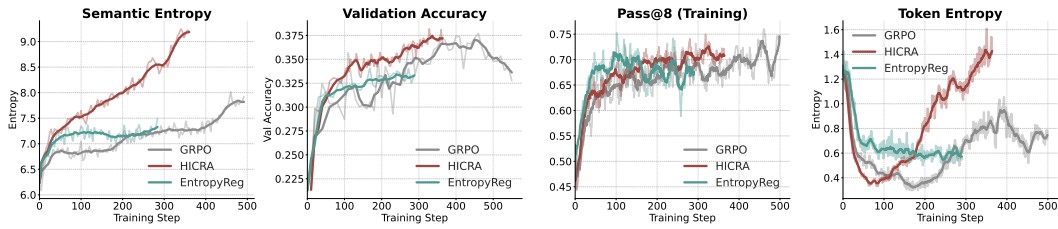

Figure 5: **HICRA vs. Entropy Regularization on Qwen2.5-7B-Base.** While entropy regularization increases token-level entropy, it fails to consistently improve accuracy and leads to uncontrolled length scaling. In contrast, HICRA boosts *semantic entropy*, which strongly correlates with validation accuracy, demonstrating the superiority of targeted strategic exploration.

To further validate this, we compared HICRA against an entropy-regularized baseline. This baseline adds (upon GRPO) an entropy regularization loss applied to all tokens uniformly. The results in Figure 5 show that promoting token-level entropy for sampling diverse tokens is counterproductive.

- The entropy regularization baseline successfully increases *Token Entropy*, but this fails to translate into performance gains; its *Validation Accuracy* stagnates and is the lowest of the three methods. This is because indiscriminately promoting token-level diversity only encourages non-productive verbosity on the vast majority of low-level tokens.

- In contrast, HICRA achieves a significantly higher *Semantic Entropy*, a targeted boost in the diversity of *strategic plans* that strongly correlates with its superior validation accuracy. This demonstrates that **the key to enhanced reasoning is not just to explore, but to focus exploration on the strategic portion of the action space.**

To validate the importance of credit assignment on the semantic level, we compare with the High-Entropy Advantage baseline, which targets high-entropy tokens as a proxy for exploration. The baseline yields competitive but inferior results compared to HICRA. As our comparison between high-entropy tokens and identified planning tokens in D.2.2 and Figure 13 shows, only 10% of high-entropy tokens serve a valid planning function. HICRA significantly outperforms entropy-based methods because it targets the semantic function of planning, rather than just statistical uncertainty.

## 5 RELATED WORK

**Reinforcement Learning for LLM Reasoning.** The application of Reinforcement Learning (RL) has been pivotal in enhancing the complex reasoning abilities of Large Language Models (LLMs). Seminal work by Ouyang et al. demonstrated the effectiveness of learning from human feedback to align models with user instructions. More recently, algorithms like Group Reward Policy Optimization (Guo et al., 2025) have been developed to specifically incentivize reasoning capabilities in LLMs,VLMs, Agents (Liu et al., 2025b; Yu et al., 2025; Team et al., 2025; Liu et al., 2025a; Wang et al., 2025c;b; Su et al., 2025; Dai et al., 2025; Zheng et al., 2025; Kang et al., 2025), leading to significant performance gains on downstream performance. While these methods have proven empirically successful, they typically apply optimization pressure agnostically across all generated tokens, without distinguishing between different functional roles within the reasoning process. Our work builds on this foundation but introduces a more targeted approach by focusing on the emergent reasoning hierarchy.

**Analysis of RL Dynamics and Exploration in LLMs.** A growing body of research seeks to understand the complex learning dynamics that occur during the RL fine-tuning of LLMs. Several studies have investigated the role of token-level entropy, observing intricate patterns and its connection to model exploration and uncertainty (Cui et al., 2025; Chen et al., 2025). Concurrently, phenomena such as sudden "aha moments" and performance improvements from longer outputs ("length-scaling") have been noted as characteristic but poorly understood outcomes of RL training (Guo et al., 2025; Liu et al., 2025b).Our paper provides a unifying framework, interpreting these phenomena as evidence of a shift from procedural learning to strategic planning.

Furthermore, recent work has identified high-entropy "fork tokens" as potential proxies for critical decision points in reasoning (Wang et al., 2025d). Our work distinguishes itself by defining planning tokens based on their semantic function. We also validate the limitation of identifying crucial tokens solely based on entropy.

Exploration-Exploitation trade-off has become a long-standing research problem in classical RL literature. Among the vast literature, Entropy Regularization or Maximum-Entropy RL (Levine, 2018; Haarnoja et al., 2017; Wang et al., 2023) is a standard technique to encourage exploration that can be seamlessly integrated with LLM RL training.

**Hierarchical Reasoning and Cognition.** The concept of hierarchical processing is a cornerstone of cognitive neuroscience, which posits that the human brain separates high-level, abstract planning from low-level motor or procedural execution (Huntenburg et al., 2018; Murray et al., 2014; Zeraati et al., 2023; Zhu et al., 2025; Xiong et al., 2025). HRM (Wang et al., 2025a) is inspired this cognitive architecture to design a specific neural architecture for hierarchical reasoning. Concurrently, this cognitive model provides a compelling parallel to the functional hierarchy we identify in RL-tuned LLMs, proposing that LLMs similarly develop a functional separation between strategic planning and procedural execution. Similar to our work, MT-Core (Pan et al., 2025) decomposed agent's behavior into coarse-grained strategic policy formulation and low-level execution, focusing on knowledge transfer in a continual learning setting. In contrast, our work concentrates on the emergent two-phase learning dynamics through RL and hierarchy-aware credit assignment for advanced LLM reasoning.

## 6 CONCLUSIONS

Our work establishes that reinforcement learning uncovers an emergent functional reasoning hierarchy in language models, demonstrating the critical performance bottleneck shifting from procedural skill to strategic exploration. This insight leads to our approach, HICRA, which demonstrates that specialized credit assignment targeting this strategic bottleneck yields more effective training. Extensive experiments validate the effectiveness of HICRA and offer deep insights into advanced reasoning through strategic exploration. Our work opens several future research directions. First, it suggests a paradigm shift away from treating all tokens equally and prompts a **rethinking of the action space** away from individual tokens toward semantic, strategic units. Second, it calls for developing **process-oriented approaches** capable of valuing correct strategic choice even if the final answer is flawed. Finally, the likely **universality of this reasoning hierarchy in complex reasoning tasks** suggests that applying these principles to domains like code generation and agentic tool-use is a valuable path forward.

ETHICS STATEMENT

All authors of this paper have read and adhere to the ICLR Code of Ethics. This work centers on the analysis of emergent hierarchical reasoning in Large Language Models (LLMs) during reinforcement learning and the development of a novel algorithm, HICRA, to improve their reasoning capabilities. Our research does not involve the use of human subjects or the collection of personally identifiable or sensitive data. The models, datasets, and benchmarks leveraged in our experiments are publicly available and have been established in prior academic work. While our research aims to advance the understanding and performance of AI reasoning systems for beneficial applications, we acknowledge that LLMs are a dual-use technology. The foundational models used in this study may reflect biases present in their original training corpora. Our work does not introduce new data sources but builds upon existing models; therefore, the potential for inherited bias is a relevant consideration for any downstream application. We have no conflicts of interest to disclose.

REPRODUCIBILITY STATEMENT

We are committed to ensuring the reproducibility of our research. All models used in our experiments, including variants of Qwen, Llama, and MIMO-VL, are open-source. The training and evaluation datasets – including DAPO, DeepScaleR, ViRL39K for training, and benchmarks like AIME, Math500, MathVista, and others for evaluation – are publicly available. A complete list is provided in Section 4.1 and Appendix D.1. The core methodology of our proposed algorithm, Hierarchy-Aware Credit Assignment (HICRA), is detailed in Section 3. A critical component of our work is the automated, data-driven pipeline for identifying "Strategic Grams" (SGs), which is fully described in Appendix A.2. To further aid replication, the complete list of SGs generated by our pipeline is provided in Listing 1. Key hyperparameters and implementation details, including the training setup and evaluation protocols, are documented in Appendix D.1. We attach the code of HICRA on top of VeRL as supplementary materials.

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

## A   EXTENDED MATERIALS OF SECTION 2

### A.1   EXAMPLE OF STRATEGIC GRAMS.

We motivate our classification of tokens by drawing a parallel to human cognition. When a person reasons through a problem, we identify their strategic thinking by its function. A phrase like, "Let's try a different approach," functions as a high-level strategic maneuver that guides the problem-solving direction. In contrast, a phrase like, "so we add 5 to both sides," is a low-level procedural step. Inspired by this functional distinction, we introduce *Strategic Grams* as a functional proxy to circumvent the difficulty of formally defining a "planning token".

---

Alright, I have this problem to solve: Given that $a$ and $d$ are ...
First, I need to minimize the expression: ...

...
Alright, so I have to find the minimum value of $S$ given these conditions.

...
Let's look at the expression:

...
I notice that both terms are fractions ...

...
Let's see if I can find relationships or substitutions. Maybe I can set $k = b + c$ and $m = a + d$. Wait, the constraint is $b + c \geq a + d$, so $k \geq m$, where $k = b + c$ and $m = a + d$. But I'm not sure if that helps directly. Alternatively, perhaps I can express $d$ in terms of other variables. Let's see.

...
Maybe I can consider specific cases or assume equality in the constraint to see what happens.

...
But wait, earlier, I assumed $b + c = a + d$. However, the given constraint ...
To confirm, I need to check if this is indeed the minimum ...

...
### Final Answer
After analyzing the problem and considering specific cases, the minimum value of the expression is:

$$\boxed{\sqrt{2} - \frac{1}{2}}$$

---

Figure 6: Reasoning from Qwen3-4B-GRPO with planning tokens (strategic grams) highlighted. Planning tokens function as the high-level strategic moves of reasoning, including logical deduction, branching and backtracing.

**Strategic Grams (SGs)** are defined as $n$-grams that function as a single semantic unit to guide the logical flow. We use n-grams because they capture the phrasal nature of strategic language (e.g., "let's consider the case") which is lost at the single-token level. These SGs facilitate three main types of logical moves: (a) deduction, (b) branching, and (c) backtracing, as we show in Figure 6.

### A.2   SG CONSTRUCTION AND SENSITIVITY ANALYSIS

A key challenge is identifying the set of SGs in a principled and reproducible manner. Manual annotation or reliance on proprietary models would introduce subjectivity and hinder reproducibility. We therefore propose an automated, data-driven pipeline based on a key insight: SGs function as the reusable **scaffolding** of a reasoning process. This function imparts a distinct statistical signature: SGs should appear frequently across a wide range of different problems but be used sparingly within any single solution. However, a significant challenge is the *linguistic diversity* of strategic language, where a single strategic intent can be expressed through numerous semantically equivalent phrases.

Our pipeline is designed to overcome these challenges by first grouping semantically equivalent n-grams and then identifying which consolidated concepts exhibit the statistical signature of strategic planning.

We construct the SG set via the following three-step procedure:

1. **Semantic Clustering:** We first extract all $n$-grams (where $n \in [3, 5]$) from a large corpus of successful reasoning solutions. Each $n$-gram is projected into a semantic embedding space using a pre-trained sentence transformer. We then apply a clustering algorithm to this embedding space. This step groups lexically diverse but semantically equivalent $n$-grams into a single cluster, directly addressing the challenge of linguistic diversity.

2. **Identification by Frequency:** To identify the clusters that represent reusable reasoning patterns, we analyze their frequency at the corpus level. For each semantic cluster, we compute its *Cluster Document Frequency (Cluster DF)*: the frequency of unique solutions that contain at least one $n$-gram from that cluster.

3. **SG Construction:** We filter for clusters with top 20% Cluster DF, implying that these SGs are common across many problems. The union of all $n$-grams within these high-frequency clusters constitutes our final set of Strategic Grams.

This pipeline result in the following collection of SGs.

Listing 1: Strategic Grams

```
{'## step', 'a good starting point is', 'a more direct approach is', 'a
    more straightforward approach',
'a simpler approach is', 'alright', 'alternatively', 'an alternative path
    is', 'an error in the thought process',
'analyze the', 'analyzing the given', 'and then find', 'another approach'
    , 'are looking for',
'based on the given', 'break down the problem', 'break it down', 'break
    it down into manageable steps',
'but', 'but wait', 'but why', 'but without more information', 'can be
    rewritten as',
'can conclude that', 'can see that', 'check if', 'consider the case where
    ', 'consider the properties of',
'correct the approach', 'define the variables', 'denote', 'determine how
    many', 'directly address the problem',
'does this hold true?', 'does this make sense?', 'double-checking the
    logic', 'finally need to',
'finally we need to', 'find a simpler', 'find a way to', 'find out how
    many', 'find the critical points',
'first need to', 'follow these steps', 'for simplicity', 'from earlier we
     have', 'from the above',
'from this, it follows that', 'from this, we can infer', 'given the
    complexity', 'given the complexity of',
'given the constraints', 'given the nature of', 'go back to the', 'goal
    is to', 'hmm,', 'hold on',
'however', 'however, we need to', 'i might have made an error', 'i need
    to re-evaluate', 'i should verify this result',
'identify the', 'identify the given information', 'if that doesn\'t work,
     we can', 'if we consider',
'in a way that', 'in the context of', 'is there a simpler method?', 'is
    there a simpler way?',
'it logically follows that', 'it seems', 'it\'s better', 'let me', 'let
    me pause and think',
'let me rethink this', 'let me verify', 'let\'s', 'let\'s analyze the
    possibilities', 'let\'s assume',
'let\'s backtrack', 'let\'s break this down', 'let\'s check our work', '
    let\'s check the constraints again',
'let\'s consider another case', 'let\'s denote', 'let\'s double-check', '
    let\'s explore a different possibility',
'let\'s formulate a plan', 'let\'s go back a step', 'let\'s outline the
    steps', 'let\'s pause and think',
'let\'s reconsider', 'let\'s try a different angle', 'let\'s validate
    this', 'looking back at the', 'maybe',
```

```
22  'maybe i can', 'my previous step was flawed', 'need to', 'need to account
        for', 'need to analyze',
23  'need to check', 'need to consider', 'need to count', 'need to determine'
        , 'need to ensure', 'need to express',
24  'need to find', 'need to follow', 'need to identify', 'need to minimize',
        'need to reconsider', 'need to show',
25  'need to solve', 'need to think about', 'need to understand', 'need to
        use', 'next', 'note that', 'now',
26  'now let', 'now need to', 'now we need to', 'okay', 'on second thought',
        'on the other hand',
27  'one way to', 'our strategy is', 'perhaps', 'perhaps i can', 'problem is
        asking', 'problem states that',
28  'proceed with the following', 'rearrange the equation', 'recall that', '
        referring to a previous step',
29  'revisiting the initial assumption', 'rewrite the equation', 'says that',
        'seems a bit complicated',
30  'should consider', 'should focus on', 'should look for', 'similarly', '
        simplify the problem', 'since',
31  'so', 'so after', 'so again', 'so the question becomes', 'so, yes', '
        something is wrong here', 'specifically',
32  'start by', 'states that', 'step by step', 'step by step reasoning', '
        step by step solution',
33  'step-by-step reasoning', 'that can\'t be right', 'that seems', 'that was
        a mistake',
34  'that assumption was incorrect', 'the correct approach is', 'the core
        idea is', 'the first step is',
35  'the key insight is', 'the key is to realize', 'the key to solving this',
        'the logical flow is',
36  'the next step is', 'the path to the solution', 'the plan is to', 'the
        problem asks for',
37  'the problem is about', 'the problem mentions', 'the problem says', 'the
        problem states',
38  'there is a mistake', 'there seems to be', 'therefore', 'think of this as
        ', 'this allows us to',
39  'this approach isn\'t working', 'this approach seems', 'this can be seen
        as', 'this implies',
40  'this implies that', 'this is because', 'this is not the correct approach
        ', 'this isn\'t leading anywhere',
41  'this leads to', 'this leads us to', 'this logically leads to', 'this
        means', 'this means that',
42  'this seems a bit', 'this suggests', 'this suggests a path', 'this
        suggests that', 'thus', 'to confirm',
43  'to consider the constraints', 'to determine', 'to do this', 'to ensure',
        'to ensure correctness',
44  'to find', 'to make it easier', 'to proceed', 'to see if', 'to solve this
        problem', 'to verify',
45  'try to', 'understand the given information', 'understanding the problem'
        , 'understanding the problem first',
46  'upon closer inspection', 'use the concept of', 'use the fact', 'use the
        fact that', 'use the method of',
47  'use the properties of', 'verify the solution', 'wait', 'wait, but', '
        wait, no', 'wait, that\'s not right',
48  'want to find', 'we are dealing with', 'we can', 'we can approach this',
        'we can conclude', 'we can deduce',
49  'we can infer', 'we can see', 'we can start by', 'we can think of this as
        ', 'we can use', 'we know',
50  'what am i missing?', 'what happens if', 'what if we assume', 'what if we
        try', 'what is being asked',
51  'which means', 'will consider the', 'work our way'}
```

This automated procedure is designed to yield a high-precision *functional proxy* for strategic planning, not an exhaustive lexicon of all possible SGs. We set reasonable hyper-parameters for identifying SGs, and we contend that the resulting SG collection is sufficiently representative to reveal the core learning dynamics. To validate this claim, we conduct a sensitivity analysis by randomly removing

30% of the identified SGs and re-running our main analysis. As shown in Figure 7 and Figure 8, the semantic entropy curves remain qualitatively identical, and the curves for perplexity and token entropy only slightly change. Semantic entropy here calculates the entropy of frequency distribution, which itself is not sensitive to slight changes in the frequency mass, as long as there are sufficient numbers of bins. This demonstrates the robustness of our SG identification and the findings derived from it.

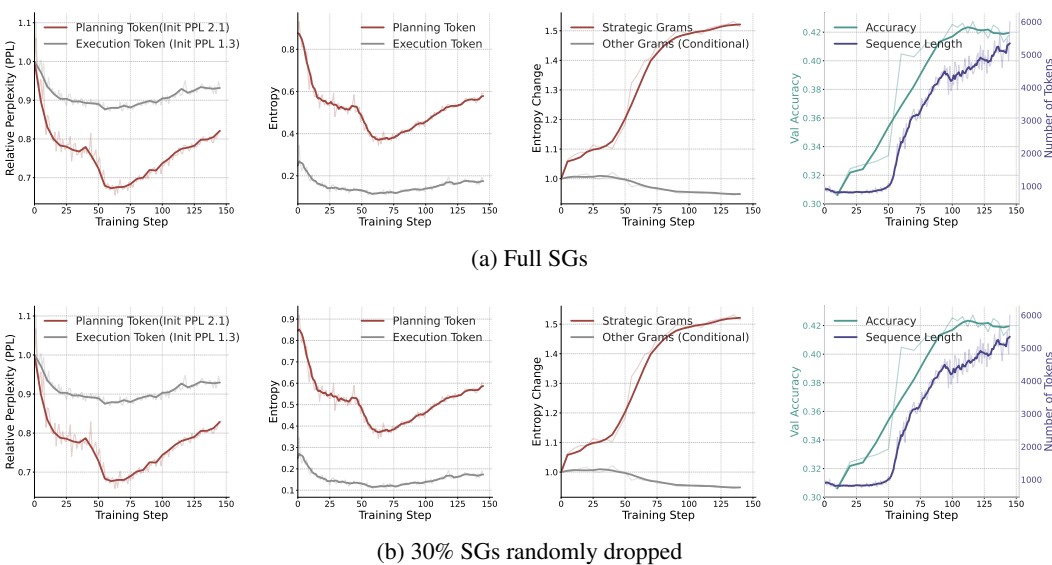

Figure 7: Sensitivity Analysis of randomly dropping 30% strategic grams on Qwen3-4B-Base training dynamics. The semantic entropy curve remain identical.

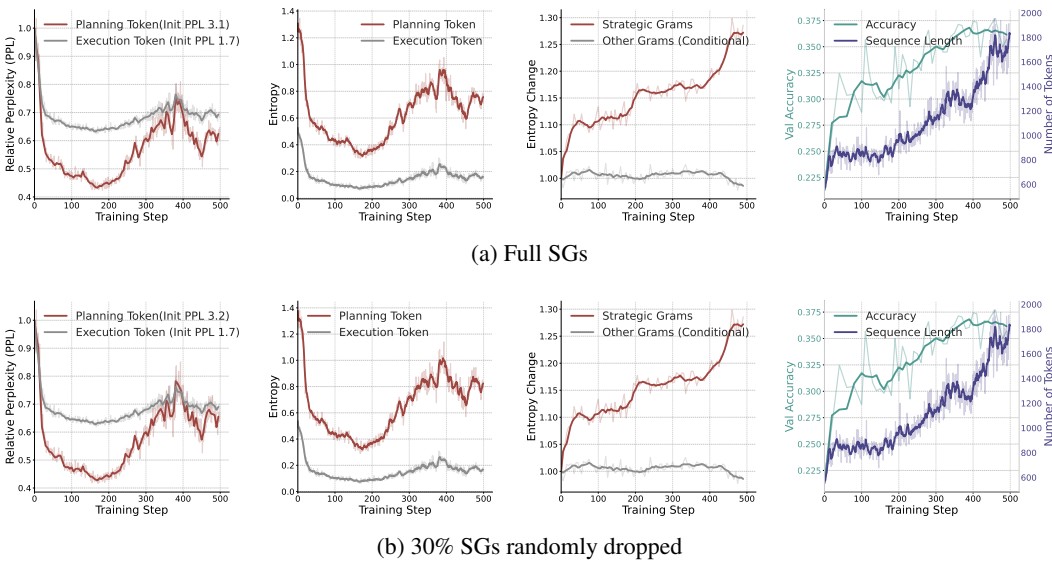

Figure 8: Sensitivity Analysis of randomly dropping 30% strategic grams on Qwen2.5-7B-Base training dynamics. The semantic entropy curve remain identical.

### A.3 HUMAN VALIDATION FOR STRATEGIC GRAMS

**Setup and Methodology** In addition to the sensitivity analysis, we also conducted a rigorous human annotation study to confirm that our pipeline identifies functional planning units with high precision.

**Setup:** We selected 250 $n$-grams that were not considered SGs extracted from training rollouts in addition to the identified SGs. **Annotators:** Three annotators were recruited from Amazon Mechanical Turk (AMT). We required Master's Qualification and explicit instructions to classify $n$-grams as "Strategic/Planning" (guiding flow, proposing plans, reflection) or "Other" (procedural, factual, formatting) given the surrounding context.

The study was designed to rigorously test the null hypothesis that the identified SGs do not serve a specialized strategic function.

**Results and Analysis.** The results confirm that our pipeline successfully filters for function. The inter-annotator agreement was substantial. However, disagreements revealed interesting nuances.

Table 2: Human Validation of Strategic Grams vs. Random N-grams

| Source List | A1 | A2 | A3 | Majority Vote |
|---|---|---|---|---|
| Identified SGs | 84% | 91% | 88% | **86%** |
| Others | 9% | 15% | 13% | **12%** |

The pipeline achieves **86**% precision for SGs (classified as strategic) vs. **12**% for random $n$-grams (classified as strategic). The distinction between the two groups is statistically stark.

**Discussion on Agreement & Disagreements:**

1. **Implicit Strategy in Random N-grams:** Humans occasionally classified random $n$-grams as "Strategic" if the context implied a heuristic leap or deduction, e.g., "Set $X = 5$". Our pipeline, which relies on frequency distribution, misses these "one-off" strategic moves.

2. **Structural Connectors in SGs:** Some high-frequency SGs were marked as "Other" by strict annotators, though they often structurally precede deduction. Despite these edge cases, the distinction between the two groups is statistically stark.

The result strongly validates the high precision of our automated SG identification pipeline, confirming that it successfully identifies linguistic units that genuinely function as the reusable scaffolding of a high-level reasoning process.

## B    FULL TRAINING DYNAMICS

Following the discussion in the main paper, we make the following further observations based on the provided training charts:

- **The initial skill-consolidation phase might be brief or absent for some models.** In the cases of the Vision-Language Models (Qwen2.5 VL-Instruct and MiMO VL-Instruct), Qwen3 4B-Instruct, Deepseek-Distill-Llama-8B, the exploration of strategic planning begins almost immediately at the start of training. This is evidenced by a significant and immediate rise in the semantic entropy of strategic grams, which occurs in tandem with a rapid boost in validation accuracy. We conjecture this is because: (a) for the VL scenarios, publicly available datasets is learned quickly by state-of-the-art models (Wang et al., 2025c); (b) strong base models like Qwen3 4B-Instruct already possess a solid foundation of low-level skills and primarily need to adapt to formatting before focusing on higher-level strategic planning.

- **Token-level entropy does not directly correlate with model accuracy.** This is strongly supported across multiple experiments. For instance, with Llama3.1 8B, Qwen3 4B-Instruct, and the VL models, token-level entropy either remains flat or decreases throughout training. During the same period, however, validation accuracy shows a steady and significant increase. This demonstrates a clear disconnect between next-token uncertainty and overall task performance.

- **Token-level entropy is misleading for policy exploration.** This observation holds true across all experiments. The Qwen3 4B-Instruct model offers a particularly stark example: its token-level entropy remains almost perfectly flat, while its semantic entropy (diversity of strategic grams) consistently increases throughout training. This contrast highlights that the variety of semantic structures a model learns is completely different from the statistical uncertainty of its next-token predictions. Figure 10 illustrates the differences of the two entropy.

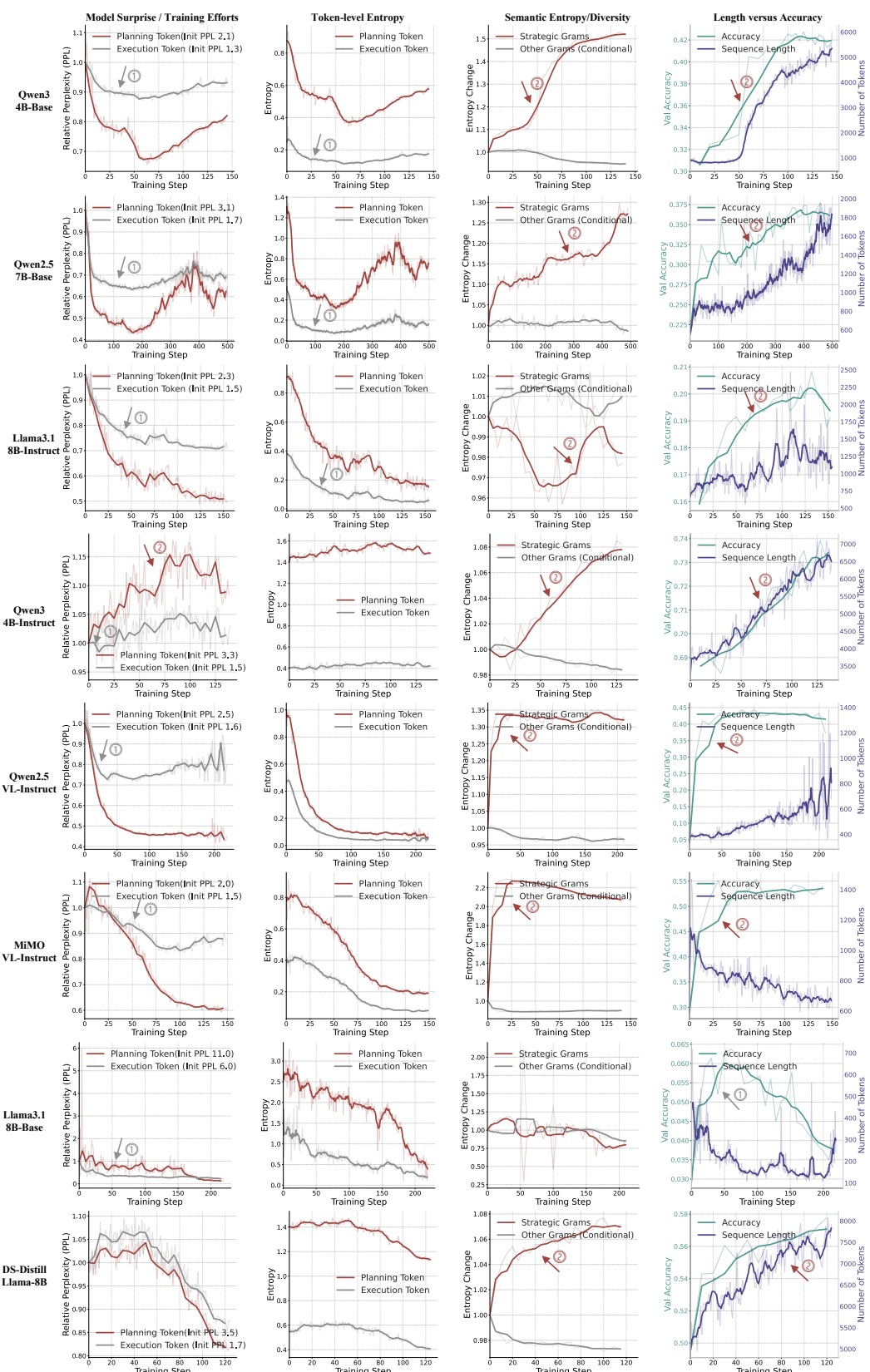

Figure 9: Training Dynamics across different LLMs and VLMs.

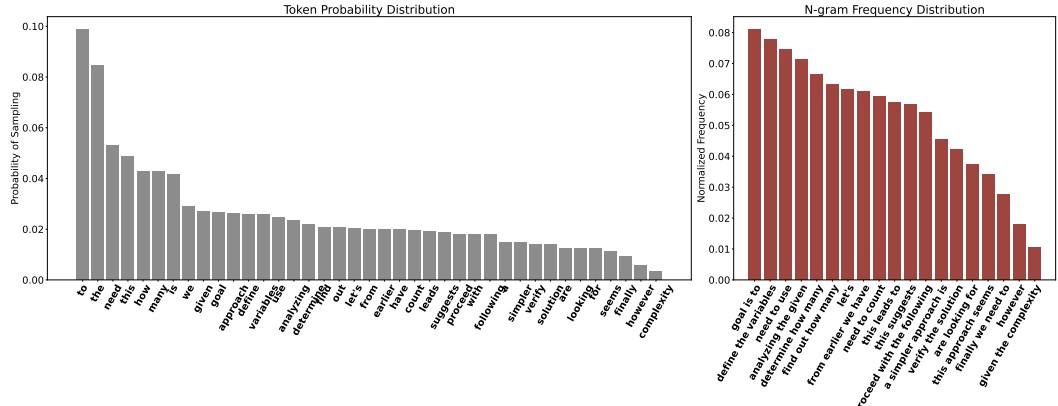

Figure 10: Comparison of Token Entropy and Semantic Entropy. (Left) Token-level Entropy is computed over the distribution of next-token probability. (Right) Semantic Entropy is computed as the Shannon Entropy over the frequency distribution of n-grams. *Intuitively, Semantic Entropy gathers tokens by their semantic function and measures the semantic diversity.* Token-level entropy is not de-duplicated by semantic meanings, and is thus dominated by vast amount of high-frequency low-level tokens.

The core difference is about scale: token-level entropy measures the uncertainty of every next-token, including the vast amount of low-level tokens such as formatting, executions that are doomed to become confident throughout training. In contrast, semantic entropy measures the diversity of the overall meanings being expressed. *A model can be very predictable in its next-token choice under a given context but still create a wide variety of different arguments or structures.*

- **Lack of Strategic Exploration Hinders Sustained Improvement in Llama Models.** We observe that the Llama-3.1-8B-Base model initially focuses almost exclusively on consolidating low-level procedural skills, a phase marked by decreasing perplexity and token entropy on execution tokens. However, once the performance gains from this procedural refinement diminish, the model fails to pivot towards exploring high-level planning strategies. This leads to performance stagnation and, eventually, degradation.

  This behavior stands in stark contrast to the more successful Deepseek Distilled Llama model, which engages in high-level strategic exploration from the very beginning of training, bypassing a distinct procedural consolidation phase. We hypothesize that for the standard Llama models, the intense initial focus on procedural correctness prematurely collapses the diversity of high-level reasoning strategies. By the time low-level skills are mastered, the model has likely converged on simpler reasoning patterns, which inhibits its ability to subsequently discover and adopt more complex and effective problem-solving approaches.

## C  THE DISTRIBUTION MATCHING PERSPECTIVE OF POLICY GRADIENTS

Imagine an ideal, or "target," policy, $\pi^*(a|s)$, that we want our current policy, $\pi_\theta(a|s)$, to emulate. We can conceptualize this target distribution as being proportional to the exponentiated advantage of the actions:

$$\pi^*(a|s) \propto \pi_{\theta_{old}}(a|s) \exp(\hat{A}(a,s)) \tag{1}$$

Or more concretely,

$$\pi^*(a|s) = \frac{1}{Z(s)} \pi_{\theta_{old}}(a|s) \exp\left(\frac{\hat{A}(a,s)}{\beta}\right) \tag{2}$$

This target policy, $\pi^*(a|s)$, re-weights the old policy based on the advantage of each action. Here, actions with a positive advantage ($\hat{A} > 0$) get their probability boosted exponentially, while actions with a negative advantage ($\hat{A} < 0$) get their probability suppressed. The term $\beta$ acts as a "temperature" parameter.

The goal is to find a new policy, $\pi_\theta$, that is as close as possible to this ideal target distribution, $\pi^*$. This is equivalent to minimizing the KL divergence:

$$\min_\theta \mathrm{KL}(\pi^*(a|s)||\pi_\theta(a|s)) \tag{3}$$

Expanding this KL divergence term:

$$\mathrm{KL}(\pi_\theta||\pi^*) = \mathbb{E}_{a \sim \pi_\theta}\left[\log \frac{\pi_\theta(a|s)}{\pi^*(a|s)}\right]$$
$$= \mathbb{E}_{a \sim \pi_\theta}[\log \pi_\theta(a|s) - \log \pi^*(a|s)]$$

Substitute our definition of $\log \pi^*(a|s) = \log \pi_{\theta_{old}}(a|s) + \frac{\hat{A}(a,s)}{\beta} - \log Z(s)$:

$$= \mathbb{E}_{a \sim \pi_\theta}\left[\log \pi_\theta(a|s) - \left(\log \pi_{\theta_{old}}(a|s) + \frac{\hat{A}(a,s)}{\beta} - \log Z(s)\right)\right]$$

$$\propto \mathbb{E}_{a \sim \pi_\theta}\left[(\log \pi_\theta(a|s) - \log \pi_{\theta_{old}}(a|s)) - \frac{\hat{A}(a,s)}{\beta}\right]$$

$$= \frac{1}{\beta}\mathbb{E}_{a \sim \pi_\theta}\left[\beta\mathrm{KL}(\pi_\theta||\pi_{\theta_{old}}) - \hat{A}(a,s)\right]$$

Minimizing this is equivalent to maximizing its negative:

$$\max_\theta \mathbb{E}_{a \sim \pi_\theta}\left[\hat{A}(a,s) - \beta\mathrm{KL}(\pi_\theta||\pi_{\theta_{old}})\right] \tag{4}$$

This expression is nearly identical to the PPO-KL objective, where the policy update is constrained using a KL divergence regularizer (Schulman et al., 2017).

Therefore, the PPO objective is essentially solving a distribution matching problem toward a target distribution shaped by the advantage function. It follows that a standard policy gradient (Williams, 2004) update nudges the policy $\pi_{\theta_{old}}$ toward an implicit target distribution $\pi^*$ defined by the advantage function. After the gradient update, the target distribution becomes the new policy sampled for exploration. Therefore, **adjusting the advantage function (or credit assignment) essentially shapes the exploration policy during RL training**.

## D    EXTENDED MATERIALS OF EXPERIMENTS

### D.1    EXPERIMENTAL SETUPS

**Training Datasets and Benchmarks**    The training dataset for LLM reasoning is sourced from DAPO (Yu et al., 2025) and DeepScaleR (Luo et al., 2025). The dataset for training VLM is sourced from ViRL39K (Wang et al., 2025c). We evaluate all models on a diverse set of challenging mathematical reasoning benchmarks to rigorously test their complex reasoning capabilities. The text-only benchmarks include AIME24, AIME25 (Mathematical Association of America, 2024), Math500 (Lightman et al., 2023), AMC23, Minerva (Lewkowycz et al., 2022), and Olympiad (He et al., 2024). We follow Deepseek R1 (Guo et al., 2025)'s evaluation protocol, where the performance is measured by Pass@1 Accuracy with temperature 0.6 sampling. For benchmarks with less than 100 queries, we use average accuracy of 32 samplings on AIME24/25 and 4 samplings on AMC23 (Yu et al., 2025). Following VL-Rethinker (Wang et al., 2025c), we evaluate on MathVista (Lu et al., 2023), MathVerse (Zhang et al., 2024), MathVision (Wang et al., 2024), and EMMA (Hao et al., 2025) for assessing multimodal reasoning across domains and disciplines. For all evaluation, we adopt strict answer matching that relies on the `\boxed` format.

**Baselines and Implementation.**    We compare HICRA against three primary baselines: the **Base** model (before RL), the widely adopted **GRPO** baseline with clip-higher (Yu et al., 2025) by default, and **Entropy Regularization**: GRPO with an additional regularization loss on token-level entropy (Cheng et al., 2025). For HICRA, we set the amplification hyperparameter $\alpha$ to 0.2 and identify planning tokens using the Strategic Grams (SGs) methodology detailed in Section 2.1. For

all experiments, we increase the training context length from 16K to 32K when the response clip rate exceeds 20% (Luo et al., 2025). For the specific experiments on Llama-3.1-Instruct, we add a dynamic filtering mechanism (Yu et al., 2025) based on GRPO Clip-Higher due to significant vanishing advantages (Wang et al., 2025c). We use two to four sets of eight A100 (80G) for training all models, and we stop the experiments if performance continues to degrade during extended training.

## D.2 ADDITIONAL ANALYSES

Table 3 shows the main results on VL models.

We also provide deeper analysis and insights into the benefits of semantic entropy for tracking exploration, differences between our approach and high-entropy tokens, and potential concerns of HICRA.

### D.2.1 SEMANTIC ENTROPY: A COMPASS FOR STRATEGIC EXPLORATION

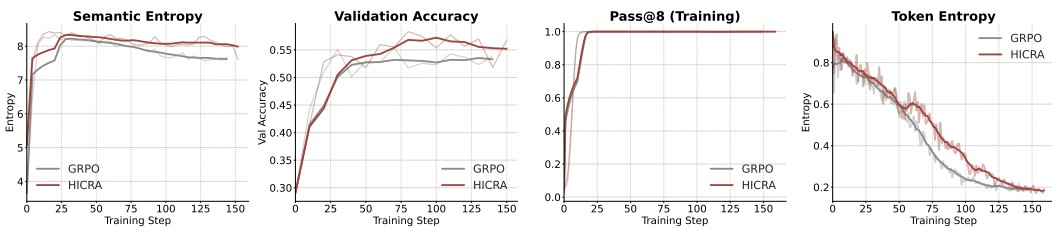

Figure 11: **Training Dynamics on MiMO-VL-Instruct-7B.** This experiment highlights that token entropy can collapse while semantic entropy remains high and predictive of validation accuracy. Furthermore, while Pass@8 saturates and is indistinguishable between methods, semantic entropy reveals a persistent exploration advantage for HICRA that translates to better final performance.

Given the crucial role of strategic exploration in unlocking reasoning performance during RL, effectively measuring it accurately is paramount. We find that semantic entropy offers distinctive benefits than common alternatives such as token-level entropy or Pass@K (Chen et al., 2021).

**Limitations of Token Entropy and Pass@K.** As shown in Figure 11 for MiMO-VL-7B, token-level entropy "collapses" for both HICRA and GRPO, simply because the vast majority of low-level tokens are doomed to become certain, thus pulling the average token entropy down. However, this decrease in token entropy might mislead researchers to suggest that exploration has ceased. Similarly, the *Pass@8 (Training)* metric quickly saturates, rendering it useless for distinguishing the ongoing learning dynamics.

**Semantic Entropy as the Differentiator.** In the same experiment, *semantic entropy* tells a more accurate story. It remains high, indicating continued exploration of diverse reasoning strategies. Crucially, HICRA consistently maintains a higher semantic entropy than GRPO, and this advantage

Table 3: Comparison of HICRA, GRPO on multimodal reasoning benchmarks.

| VLM | MathVista | MathVision | MathVerse | EMMA |
|---|---|---|---|---|
| **MiMO-VL-Instruct-2508** | | | | |
| Base | 77.0 | 42.9 | 61.8 | 36.3 |
| GRPO | 73.7 | 42.8 | 63.0 | 41.9 |
| HICRA | **80.7** | **48.9** | **65.4** | **44.1** |
| Δ (HICRA - GRPO) | +7.0 | +6.1 | +2.4 | +2.2 |
| **Qwen2.5-VL-7B-Instruct** | | | | |
| Base | 66.6 | 23.6 | 45.9 | 22.3 |
| GRPO | 70.8 | 25.8 | 48.8 | 31.8 |
| HICRA | **71.4** | **28.7** | 48.2 | **33.0** |
| Δ (HICRA - GRPO) | +0.6 | +2.9 | -0.6 | +1.2 |

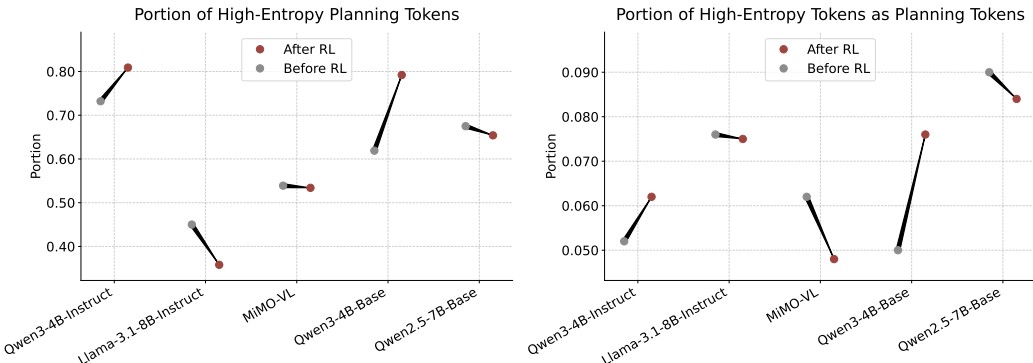

Figure 12: **Planning Tokens vs. High-Entropy Tokens.** (Left) A majority of our functionally-defined planning tokens are also high-entropy (top 30%). (Right) However, the reverse is not true; most high-entropy tokens are not planning tokens.

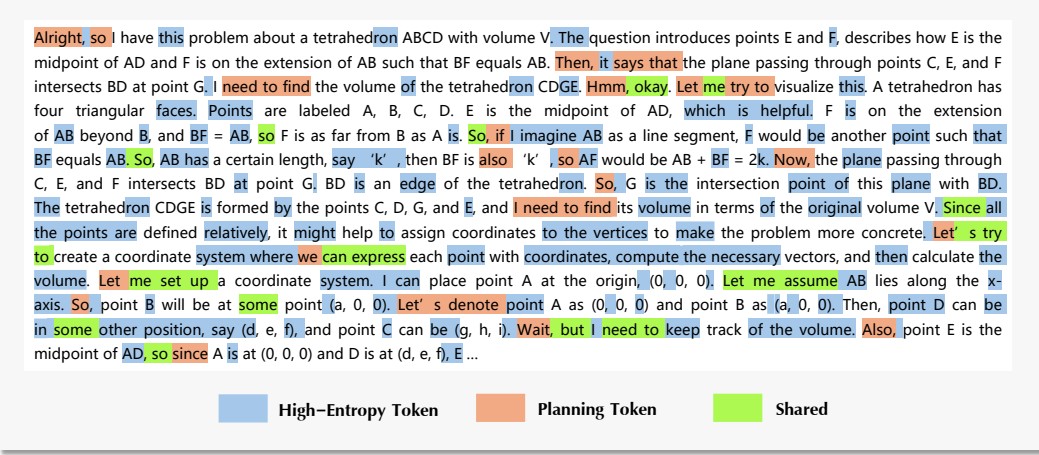

Figure 13: Planning Tokens, High-Entropy Tokens and Shared Tokens are highlighted with different colors. This concrete example suggests how these two definitions differ: Planning Tokens function as strategic skeletons of a reasoning solution and are thus sparse, with more than half of these semantic units also having higher entropy. In contrast, a majority of high-entropy tokens only exhibits high-variations in its phrasing, spreading across low-level executions and high-level planning. Fig. 12 reveals that less than 10% high-entropy tokens serve the semantic function of planning.

directly correlates with its superior final validation accuracy. This also demonstrates the generality of our approach, extending effectively to multimodal reasoning tasks on vision-language models like MiMO-VL-7B.

### D.2.2 PLANNING TOKENS VS. HIGH-ENTROPY "FORK" TOKENS

Recent work has proposed high-entropy tokens, sometimes called "fork tokens," to imply its role as proxies for decision points in a reasoning trace (Wang et al., 2025d). Our analysis investigates the relationship between our functionally-defined planning tokens and this entropy-based definition.

Figure 12 reveals a crucial asymmetry. While a majority of planning tokens exhibit high entropy (aligning with their role as points of strategic choice), the reverse is not true: most high-entropy tokens are *not* planning tokens. This finding highlights the limitations of using high entropy as a standalone proxy for strategic function. High token-level entropy ensures sampling diversity, but it does not guarantee semantic function. **Many high-entropy tokens may correspond to variations in phrasing or calculation that do not alter the high-level reasoning path.** In contrast, our approach identifies tokens based on their functional role in orchestrating the solution, providing a more direct and reliable signal for strategic credit assignment.

### D.2.3 BOUNDARY CONDITIONS OF HICRA

Our hierarchical framework predicts that strategic exploration is only beneficial once a foundational level of procedural competence is established. HICRA's effectiveness is predicated on a key assumption: that the base model should readily possess a reasonable foundation for low-level procedural correctness. As shown in Figure 14, when this foundation is lacking – as observed with Llama-3.1-Instruct – HICRA can fail to provide an advantage over GRPO. Seen from the semantic entropy graph, there is a reverse trend between GRPO and HICRA, implying an opposite training focus on planning tokens and execution tokens. HICRA's enforced strategic exploration becomes counterproductive if the model cannot reliably execute the plans it generates, leading to unstable learning dynamics and learning effects observed on Llama-3.1. This delineates the scope of HICRA, suggesting that HICRA is most effective when applied to models that have already achieved a degree of procedural reliability, highlighting an important dependency for future work on more adaptive, model-aware hierarchical methods.

Figure 14: HICRA on Llama-3.1-Instruct-8B.

## E LLM USAGE

During the preparation of this manuscript, the authors utilized a large language model (LLM) for assistance with editing. The LLM was used to improve grammar, clarity, and the overall logical flow of the text. However, the core scientific contributions, including the conceptualization of the research, experimental design, analysis, and the final conclusions, are entirely the work of the human authors.

