# OpenReview forum: "Emergent Hierarchical Reasoning in LLMs through Reinforcement Learning"
_ICLR.cc/2026/Conference — ICLR 2026 Poster_

### Official Review · Reviewer_RzuK · 2025-10-30

**Soundness:** 3
**Presentation:** 3
**Contribution:** 3
**Rating:** 6
**Confidence:** 4

**Summary:**

This paper investigates the learning dynamics of Large Language Models during reinforcement learning for reasoning tasks. The authors propose a two-phase learning process. Models first consolidate low-level procedural skills and then shift to exploring high-level strategic planning. This paper argues that the second phase is the main bottleneck for performance. Based on this insight, the authors introduce HICRA, a new algorithm that modifies the credit assignment in RL. HICRA focuses optimization pressure on "planning tokens", which guide the reasoning process. Experiments across multiple models and benchmarks show that HICRA outperforms standard RL baselines by accelerating strategic exploration.

**Strengths:**

1. The paper presents a clear and insightful analysis of the learning process in LLMs. The proposed two-phase framework provides a compelling explanation for several previously observed but poorly understood phenomena, such as "aha moments" and "length scaling". This provides a valuable conceptual model for the community.

2. The proposed method, HICRA, is simple, well-motivated, and directly addresses the bottleneck identified in the analysis. Instead of introducing a complex new architecture, it makes a targeted and intuitive change to the advantage function of an existing algorithm, making it easy to understand and implement.

3. The empirical evaluation is extensive and convincing. The authors validate their claims across a diverse set of open-source language and vision language models on multiple challenging mathematical reasoning benchmarks.

**Weaknesses:**

1. The method's effectiveness is dependent on the quality of the "Strategic Grams" used to identify planning tokens. The automated pipeline for extracting these grams seems heuristic and may not be robust. It is unclear if it truly identifies strategic reasoning or just common, high-frequency phrases that correlate with correct answers.

2. The set of compared baselines could be stronger. The paper argues that its functional definition of planning tokens is superior to using high-entropy "fork tokens" as a proxy. However, it does not include a direct experimental comparison to a baseline that selectively rewards high entropy tokens. Such a comparison is necessary to validate the additional complexity of the proposed semantic identification pipeline.

3. The paper frames the problem with a binary distinction between planning and execution tokens. However, this is a simplification of complex reasoning, which can involve multiple layers of strategic abstraction. The method treats all planning tokens equally, which might not be optimal for learning intricate, multi-step plans.

4. The method's applicability appears to be limited by the base model's initial capabilities. The appendix shows that HICRA can perform worse than the baseline on models that lack a solid procedural foundation. This suggests the method is not a general solution but rather an accelerator for already competent models.

**Questions:**

1. The identification of Strategic Grams is a critical component. How sensitive is HICRA's performance to this list? For example, what is the impact of using a much smaller list, or a list generated from a different data domain? This would help clarify whether the method is learning general planning skills or overfitting to certain phrases.

2. Could the authors provide a direct experimental comparison against a baseline that uses a high-entropy heuristic to identify and reward important tokens?

3. Could the authors assign different weights to different types of planning tokens, distinguishing between high-level strategy and lower-level tactical steps?

---

> ### Author Response · Authors · 2025-11-22
> **Author Rebuttal**
>
> # Response to Reviewer RzuK
> We sincerely thank you for your positive assessment and for recognizing our work as a valuable conceptual model that offers a compelling explanation for puzzling RL phenomena. We appreciate your constructive feedback on the robustness of Strategic Grams and the need for stronger baselines.
>
> **We kindly ask you to refer to our General Response at the beginning of the thread. In particular, we highlight the section Catalyzing Future Research**, which outlines how our Emergent Reasoning Hierarchy framework benefits the community by shifting focus from token-level to semantic-level RL and guiding the design of adaptive algorithms.
>
> Below, we address your specific questions, incorporating new experiments and analyses presented in the General Response.
> ***
> ### **1. Comparison to High-Entropy Baselines (Response to Q2)**
>
> You asked for a direct experimental comparison against a baseline that selectively rewards high-entropy tokens to validate the necessity of our semantic pipeline. **We implemented this baseline (High-Entropy Advantage) and present the results in the table in the General Response.** This result also aligns with existing work.
>
> The High-Entropy Advantage baseline failed to achieve significant gains over the standard GRPO baseline. This result is directly explainable by our analysis in Appendix D.2.2: while planning tokens often have high entropy, the vast majority of high-entropy tokens (over 90%) are not planning tokens. They are often trivial variations in wording or syntax. Therefore, blindly rewarding high entropy tokens introduces significant noise.
>
> This experiment confirms that the semantic identification provided by HICRA is necessary. The performance gain comes from targeting the functional role of planning, not just statistical uncertainty.
> ***
> ### **2. Robustness of Strategic Grams (Response to Q1)**
>
> You asked about the sensitivity of HICRA to the list of Strategic Grams (SGs) and whether it might be overfitting to specific phrases. **We addressed this with three complementary analyses:**
>
> First, as shown in existing analysis of Appendix A.2 (Fig 7 & 8), randomly dropping 30% of SGs did not qualitatively alter the learning dynamics or performance. This indicates the method relies on a dense, redundant signal of strategic planning rather than a specific magic list.
>
> Second, in the General Response, we report a Placebo Test where we applied HICRA’s credit boost to random n-grams. This resulted in negative performance gains. This proves that HICRA is not overfitting to arbitrary phrases; the improvement is driven specifically by the semantic signal of planning tokens.
>
> Third, our human evaluation confirmed that 86% of identified SGs are functionally pertaining to strategic planning.
> ***
> ### **3. The Binary Distinction (Response to Q3)**
>
> You noted that the binary distinction between planning and execution is a simplification of complex reasoning. We agree, but **we wish to clarify that our model does not imply a strict sequential separation where learning phases are decoupled or mutually exclusive.**
>
> As explicitly stated in the paper (Line 162), the learning process exhibits overlapping phases; our analysis (Figure 2) shows that the model continues to refine execution (lowering procedural entropy) even as strategic exploration begins. Our claim is that the primary bottleneck for performance shifts: Phase 1 focuses on consolidating procedural reliability, while Phase 2 shifts the frontier of improvement to strategic planning. These phases are distinct in their dominance of the learning signal, but we do not claim there is a strict sequential separation.
>
> That said, we agree that your suggestion to assign different weights to distinct types of planning is an excellent direction for future work to capture more granularity. This aligns with our vision for Adaptive RL Methods, where algorithms dynamically adjust credit based on the specific type of reasoning bottleneck encountered.
> ***
> ### **4. Dependency on Base Model Capabilities (Response to Weakness 4)**
>
> You correctly noted that HICRA yields no gain on models without a solid procedural foundation. **As we documented and analyzed in D.2.3 and appendix B, this is a strong empirical validation of our theoretical framework rather than a methodological failure.** Our theory posits that Strategic Exploration becomes the primary bottleneck after Procedural Competence is established. The fact that HICRA does not help a model that struggles with low-level defines the boundary conditions of the method and validates the hierarchical nature of reasoning.
> ***
> ### **Final Appeal**
>
> We believe we have effectively addressed your concerns. Given that our work provides a valuable conceptual model for the community with extensive empirical validation and opens up promising directions as noted in the general response, we sincerely ask that you could consider raising your score to support the acceptance of this work.

---

> > ### Comment · Reviewer_RzuK · 2025-11-27
> >
> > Thanks for conducting the additional High-Entropy Advantage baseline and Placebo Test experiments during the rebuttal.
> >
> > My concerns are well-addressed. I will keep my score unchanged. I hope the authors can consider incorporating these results into the final manuscript.

---

> > > ### Author Response · Authors · 2025-11-27
> > > **Thank you**
> > >
> > > Dear Reviewer RzuK,
> > >
> > > Thank you for your time, efforts and prompt replies. We have now updated the manuscript, incorporating the clarifications, new experiments.
> > >
> > > Thank you once again for your positive assessment. **We sincerely invite you to review the "Catalyzing Future Research" section in our General Response, which we believe benefits broad audience in the field including the reviewers.** We would also be more than appreciated if you could raise your score.
> > >
> > > Best Regards, The Authors

---

### Official Review · Reviewer_QnoB · 2025-10-30

**Soundness:** 2
**Presentation:** 3
**Contribution:** 2
**Rating:** 4
**Confidence:** 3

**Summary:**

This paper presents an analysis of RL dynamics in large language models, proposing that enhanced reasoning emerges from a hierarchical separation between high-level strategic planning and low-level procedural execution. The authors introduce a data-driven method to identify "planning tokens" and build upon this insight with HICRA, a targeted credit assignment algorithm that consistently outperforms strong GRPO baselines across various models and mathematical reasoning benchmarks. This demonstrates that their conceptual framework is not merely theoretical but offers a practical and effective lever for improving model performance.

**Strengths:**

1. The paper's focus on identifying and boosting "reasoning" or "planning" tokens is an interesting motivation. Experiments demonstrates that this is a practical lever for improving model performance.

2. Its proposed algorithm, HICRA, demonstrates consistent performance gains across several small models and benchmarks, showing the effectiveness and generality of the approach.

**Weaknesses:**

1. The notion of a decisive shift and strict sequential separation (first one, then the other) may over-simplify the RL training process in a massive over-parameterized model like an LLM. In reality, planning and execution are rarely decoupled in learning. Improvements in low-level procedural skills (e.g., generating syntactically correct code or flawless calculation steps) immediately inform and enable more ambitious, complex, and high-level strategic plans. Conversely, a good high-level plan (strategic reasoning) acts as a powerful curriculum for focusing the procedural execution component, leading to faster skill acquisition.

2. The proposed solution, Hierarchy-Aware Credit Assignment (HICRA), relies heavily on the ability to consistently identify and isolate "high-impact planning tokens" from procedural execution tokens. However, ground-truth labels for token function are lacking in this field. Since LLMs generate text in an autoregressive fashion, the designation of a token as "planning" or "procedural" is not externally supervised. It must be determined by some internal, emergent mechanism (e.g., attention head patterns, internal self-correction signals, or specific activation clusters). If HICRA relies on a learned or heuristic mechanism to identify the hierarchy, the success of the credit assignment becomes critically dependent on the robustness and stability of that internal mechanism. A noisy or incorrect classification of a "planning" token as "procedural" will cause the reward signal to be diluted (the problem HICRA aims to solve). Conversely, misclassifying a "procedural" token as "planning" could lead to unstable and high-variance gradients in the early layers responsible for high-level reasoning, potentially destabilizing the planning component itself.

3. While concentrating credit on planning tokens targets the apparent bottleneck, it simultaneously starves the procedural components of the direct, high-magnitude signal they need to maintain or improve their correctness. For complex, multi-step tasks, procedural fidelity is non-negotiable. Reducing the optimization pressure on these steps may lead to a model that has great plans but consistently fails on the execution details.

4. This work builds heavily upon the intersection of methodologies from at least three areas: hierarchical RL for LLM agents with explicit planning modules; fine-grained credit assignment \& token-specific optimization; RL training dynamics. This makes it difficult to identify a significant contribution of this paper. For example, MT-Core [1] also consists of a coarse-grained policy formulation and a fine-grained policy learning by LLM and RL, respectively. Then what is the key contribution of this paper with respect to the methodologies from these three areas?

[1] Pan, C., Yang, X., Wang, H., Wei, W., \& Li, T. (2024). Hierarchical continual reinforcement learning via large language model. arXiv preprint arXiv:2401.15098.

5. The core limitation lies in the algorithm's dependency on a foundational level of procedural correctness. HICRA's method of selectively amplifying the signal for high-level planning tokens is most effective *after* the base model has achieved a certain level of procedural reliability. If the base model lacks these reliable low-level skills, the enforced strategic exploration can be counterproductive, as a brilliant strategy is useless if the execution (e.g., a simple calculation) is consistently flawed. More experiments should be conducted to sufficiently  explore the performance of HICRA on models with weak initial procedural skills, such as MiMO-VL-Base.

The current form of the paper is not enough convincing from motivation to methodology.

**Questions:**

See weaknesses.

---

> ### Author Response · Authors · 2025-11-22
> **Author Rebuttal**
>
> # Response to Reviewer QnoB
>
> We thank you for your review and for recognizing that our proposed algorithm, HICRA, serves as a **"practical and effective lever"** for improving model performance. We appreciate your engagement with the work.
>
> We respectfully request that you review the General Response regarding the clarification on the core contribution and promising future directions. We noticed that significant attention in the review was placed on the HICRA algorithm as a standalone method. However, **we wish to clarify that the primary contribution of this paper is the Emergent Reasoning Hierarchy, extending beyond a single RL algorithm.**  HICRA serves primarily as a test and a example to validate this theory and showcase how we can design RL algorithm based on the framework.
>
> ***
> ### **1. Clarification on "Strict Sequential Separation" (Weakness 1)**
> You suggest that our "two-phase" model implies a strict sequential separation that over-simplifies the learning process.
>
> We wish to clarify that we **do not claim learning phases are decoupled or mutually exclusive.** As explicitly stated in the paper, the learning process exhibits overlapping phases (in Line162). Our analysis (Figure 2) shows that the model continues to refine execution (lowering procedural entropy) even as strategic exploration begins.
>
> Our claim is that the primary bottleneck for performance shifts. Phase 1 focuses on consolidating procedural reliability, while Phase 2 shifts the frontier of improvement to strategic planning. These are distinct in their dominance of the learning signal, but we do not claim there is a sequential separation.
> ***
> ### **2. Reframing the "Failure" on Weak Models (Weakness 3 & 5)**
>
> You raise concerns that HICRA might "starve" procedural components and noted its poor performance on weak models like `Llama-3.1`.
>
> We wish the reviewer won't penalize the paper for a transparently documented boundary condition that actually validates our core theory.
>
> * **Success on 6 out of 7 Models:** We extensively tested HICRA across seven different base models. HICRA achieved consistent, significant improvements on six of them. This demonstrates that the method is robust and generalizable across the vast majority of standard setups.
> * **Llama-3.1 Validates the Theory:** The lack of improvement on `Llama-3.1` was fully analyzed in **Appendix D.2.3**. Our theory explicitly predicts that Strategic Exploration can only yield gains if Procedural Competence) is established. `Llama-3.1` lacked this foundational competence. Therefore, its failure to improve is not a flaw in the algorithm, but strong empirical confirmation of the Hierarchical Reasoning framework.
> * **Procedural Fidelity:** We agree that procedural fidelity is non-negotiable. We wish to clarify that we don't claim it is unimportant; rather, **we explicitly state the boundary condition for HICRA in appendix D.2.3**, that "HICRA’s effectiveness is predicated on a key assumption: that the base model should readily possess a reasonable foundation for low-level procedural correctness", this highlights procedural fidelity as the foundation. We wish the reviewer does not penalize the paper for what it clearly documented and analyzed.
> ***
> ### **3. Distinction from Hierarchical RL (Weakness 4)**
>
> You asked for the contribution relative to Hierarchical RL (HRL) like MT-Core.
>
> Our contribution is fundamentally distinct from standard HRL:
> * **Emergent vs. Architectural:** "Hierarchical RL" typically relies on architectures with explicit, separate modules or policies (coarse-grained vs. fine-grained). In contrast, our "Emergent Reasoning Hierarchy" describes a functional phenomenon arising within a standard, single-policy language model trained via RL.
> * **Methodological Innovation:** We do not require complex architectural changes. Instead, we introduce HICRA, demonstrating that
> we can induce hierarchical reasoning behaviors simply by reshaping the gradient based on semantic function on the tokens.
> ***
> ### **Final Appeal**
>
> We respectfully argue that the "weakness" regarding `Llama-3.1` should be viewed as a scientifically rigorous analysis of boundary conditions, not a failure of the method. Given that
> - our framework provides a unifying theory for understanding and designing reasoning RL algorithms
> - HICRA delivers consistent gains on 6 other major models and we carefully analyzed the boundary conditions
> - the paper fuels meaningful research in the community as noted in the General Response
>
> we believe the contribution significantly outweighs the limitations if any. We respectfully ask you to **re-evaluate the paper and consider raising your score** to support paper acceptance.

---

> ### Author Response · Authors · 2025-11-27
> **Awaiting Response**
>
> Dear Reviewer QnoB,
>
> As the discussion period draws to a close, we kindly invite you to check our response and the updated manuscript.
>
> We have put substantial effort into addressing your concerns. We included clarifications, new experimental results (two baselines) and human validation on SGs to strengthen the claims made in the paper. In addition, we appreciate you bringing MT-Core to our attention and we include a discussion with MT-Core in the related work section.
>
> **We also invite you to review the "Catalyzing Future Research" section in our General Response, which we believe benefits broad audience in the field including the reviewers**. We believe our framework -- the Emergent Reasoning Hierarchy -- provides a critical shift from token-level to semantic-level RL, offering a unified theory that benefits the broader community and future research in the field.
>
> **Given the significant value of this work and our clarifications regarding the method's theoretical soundness, we respectfully ask that you consider raising your score to support the acceptance of this work.**
>
> Best regards,
> The Authors

---

> ### Author Response · Authors · 2025-11-28
> **Awaiting Response**
>
> Dear Reviewer QnoB,
>
> As the discussion period draws to a close, we kindly invite you to check our response and the updated manuscript.
>
> We have put substantial effort into addressing your concerns. We included clarifications, new experimental results (two baselines) and human validation on SGs to strengthen the claims made in the paper.
>
> **We also invite you to review the "Catalyzing Future Research" section in our General Response, which we believe benefits broad audience in the field including the reviewers**. We believe our framework -- the Emergent Reasoning Hierarchy -- provides a critical shift from token-level to semantic-level RL, offering a unified theory that benefits the broader community and future research in the field.
>
> **Given the significant value of this work and our clarifications regarding the method's theoretical soundness, we respectfully ask that you consider raising your score to support the acceptance of this work.**
>
> Best regards,
> The Authors

---

### Official Review · Reviewer_2wpP · 2025-10-31

**Soundness:** 2
**Presentation:** 3
**Contribution:** 2
**Rating:** 4
**Confidence:** 4

**Summary:**

The paper studies why RL improves LLM reasoning and proposes that training exhibits an emergent two-phase hierarchy: (1) early consolidation of low-level procedural execution (formatting, arithmetic), evidenced by reduced perplexity and token entropy for “execution tokens”; (2) later exploration and mastery of high-level strategic planning, evidenced by increased “semantic entropy” of “Strategic Grams” (SGs)—n-gram phrases that purportedly carry strategic functions (deduction, branching, backtracing). The authors claim this framework unifies puzzling observations (“aha moments,” length-scaling, token entropy dynamics). They argue that standard GRPO distributes optimization uniformly across tokens, diluting credit on high-impact planning tokens. They propose HICRA, a simple GRPO modification that amplifies token-level advantages for planning tokens (identified by SGs) and dampens their penalties, thereby focusing optimization on strategy. Across several LLMs and VLMs on math reasoning benchmarks (e.g., AIME24/25, Math500, AMC23, Minerva, Olympiad; MathVista, MathVerse, MathVision, EMMA), HICRA reportedly outperforms GRPO baselines. They further analyze error-type dynamics (planning vs. others), show semantic entropy is a better exploration indicator than token entropy or Pass@K, and discuss limitations (e.g., dependence on a solid procedural base; weaker gains on Llama-3.1-8B-Instruct).

**Strengths:**

1. Clear, testable hypothesis about a two-phase, hierarchical learning dynamic with consistent qualitative evidence across multiple model families (text-only and VLM).

2. A simple, implementable algorithm (HICRA) grounded in the analysis; integrates cleanly with GRPO and yields consistent improvements on several benchmarks.

3. Useful analytic tools: semantic entropy over strategic units; error-type diagnostics; sensitivity analyses showing robustness of qualitative trends to partial SG removal.

4. Solid empirical sweep: multiple models, several benchmarks, process metrics (PPL/entropy/length/accuracy).

**Weaknesses:**

1. Operationalization of “planning tokens” via SGs is heuristic and potentially brittle: depends on corpus choice, clustering hyperparameters, n-gram length range, and English phrasing. No human validation statistics (precision/recall) against expert annotations; no cross-domain generalization test (beyond math/VLM math).

2. Causal claims remain tentative: the evidence is largely correlational. In particular, the “two-phase” dynamic and the linkage between semantic entropy and accuracy could be confounded by curriculum effects, context-length growth, reward clipping, or prompt-format shifts.

3. Dependence on pre-existing procedural competence: HICRA underperforms/struggles on weaker models (e.g., Llama-3.1-8B-Instruct), suggesting the method is not broadly robust. No adaptive mechanism to detect when to emphasize execution vs. planning.

4. Reward shaping risk: amplifying credit on planning tokens may overfit to stylistic strategic phrases (SG lexicon) rather than genuine strategy. The overlap/shift of SGs across datasets or languages is unclear; multilingual robustness is untested.

5. Evaluation transparency concerns: many datasets are small and sensitive; multiple sampling runs, seeds, and statistical tests are not fully reported. Some deltas are modest; one negative result (Olympiad on Qwen3-4B-Instruct) is noted.

6. Comparison set: strong recent baselines (e.g., process-reward shaping, reflection-based RL, decision-token methods) are not exhaustively compared; entropy-regularization is a weak foil.

**Questions:**

1. Can you provide quantitative validation of SG detection (human annotation study with precision/recall; inter-annotator agreement)? How sensitive are results to different embedding models, clustering methods, n-gram ranges, and thresholds?

2. Can you run targeted causal tests—e.g., (i) randomly reassign SG labels to matched-frequency n-grams; (ii) ablate SG presence at train time (mask or paraphrase them away); (iii) directly reward semantic-entropy increases vs. HICRA—to test whether planning-focus per se drives gains?

3. Can HICRA dynamically down-weight planning tokens when execution errors dominate (e.g., via online estimation of error types) to avoid failures on weaker models?

4. Generalization: Do the findings hold on non-math reasoning (commonsense, code), agentic tool-use, and multilingual tasks? How portable is the SG lexicon across domains and languages?

5. Robustness and stats: Please report means/standard deviations over multiple seeds for key benchmarks; include significance tests. Are improvements consistent under stricter decoding (temp=0) or chain-of-thought suppression?

6. If you paraphrase SG phrases (same function, different wording), do gains persist? What fraction of HICRA’s advantage survives when SG phrases are held out from training prompts?

7. Does HICRA exacerbate verbosity or hallucinations in non-math tasks? Provide length-normalized metrics and calibration analyses.

---

> ### Author Response · Authors · 2025-11-22
> **Author Rebuttal (Part 1)**
>
> # Response to Reviewer 2wpP
>
> We sincerely thank you for your detailed and constructive review. We appreciate that you found our hypothesis **"clear and testable"** and our empirical sweep **"solid."**
>
> We respectfully request that you review the General Response regarding the clarification on the core contribution and promising future directions. We noticed that significant attention in the review was placed on the HICRA algorithm as a standalone method. However, **we wish to clarify that the primary contribution of this paper is the Emergent Reasoning Hierarchy, extending beyond a single RL algorithm.**  HICRA serves primarily as a test and a example to validate this theory and showcase how we can design RL algorithm based on the framework.
>
> Due to the limited rebuttal period, we prioritized experiments that challenge the validity of this framework (e.g., "Is the hierarchy real?" "Is the causality valid?"). We address your specific requests below.
>
> ### **1. Validating "Strategic Grams": Heuristics vs. Function (Addressing Q1)**
>
> To address your concern regarding the heuristic nature of Strategic Grams (SGs), we conducted a rigorous human annotation study which confirms that our pipeline identifies functional planning units with high precision.
>
> * **Setup:** We select 250 n-grams that are not considered SGs extracted from training rollouts in addition to the identified SGs.
> * **Annotators:** Three annotators were recruited from Amazon Mechanical Turk (AMT). We required Master's Qualification and explicit instructions to classify n-grams as "Strategic/Planning" (guiding flow, proposing plans, reflection) or "Other" (procedural, factual, formatting) given the surrounding context.
>
> **Table 1: Human Validation of Strategic Grams vs. Random N-grams**
>
> | Source List | Annotator 1 (% Strategic) | Annotator 2 (% Strategic) | Annotator 3 (% Strategic) | **Majority Vote (% Strategic)** |
> | :--- | :---: | :---: | :---: | :---: |
> | **Identified SGs** | 84% | 91% | 88% | **86%** |
> | **Others** | 9% | 15% | 13% | **12%** |
>
> **Discussion on Agreement & Disagreements:**
> The results confirm that our pipeline successfully filters for function (**86% precision** for SGs vs **12%** for random n-grams). The inter-annotator agreement was substantial. However, disagreements revealed interesting nuances:
> 1.  **Implicit Strategy in Random N-grams:** Humans occasionally classified random n-grams as "Strategic" if the context implied a heuristic leap or deduction, e.g., "Set x=5". Our pipeline, which relies on frequency distribution, misses these "one-off" strategic moves.
> 2.  **Structural Connectors in SGs:** Some high-frequency SGs were marked as "Other" by strict annotators, though they often structurally precede deduction.
> Despite these edge cases, the distinction between the two groups is statistically stark.
>
> ### **2. Establishing Causality: The "Placebo" Experiment (Addressing Q2)**
>
> To distinguish between genuine planning improvements and simple reward shaping, we performed a "Placebo HICRA" experiment on Qwen2.5-7B-Base that isolates the causal impact of semantic function.
> We replaced the verified SGs with the set of random n-grams and applied the same HICRA credit amplification. The result is shown in the table in the General Response. The results show a performance decrease, providing a causal evidence that the performance boost is driven by the **semantic function** of the planning tokens.
>
> ### **3. Baseline for "Semantic Entropy" Optimization (Addressing Q4)**
>
> Regarding the comparison to a direct "Semantic Entropy" optimization baseline, we clarify that the metric's distributional nature makes it suitable as a diagnostic tool rather than a token-level reward.
>
> Semantic Entropy defined in our paper in Section 2.2 (also see the equation in 4.), is a distributional metric computed at the **aggregate level** for each training iteration (across the batch), **not at the granularity of a single response, making it impossible to be used as a individual reward signal"**. Its purpose is to serve as a diagnostic tool for tracking the diversity of strategic n-grams throughout training, not as an individual reward signal.
>
> Because it is an iteration-level statistic, it is methodologically impossible to use it as a direct reward signal for individual trajectories in a standard RL framework. This is why we compared against Entropy Regularization (Figure 5), which *is* the standard method for encouraging exploration, and showed that HICRA outperforms it. We also add a new baseline "High-Entropy Advantage", as shown in the table in the General Response

---

> > ### Author Response · Authors · 2025-11-22
> > **Author Rebuttal (Part 2)**
> >
> > ### **4. Re-evaluating the "Failure" on Weak Models (Addressing Q3)**
> >
> > Regarding the performance on weaker models and the potential for dynamic weighting, we argue that the observed limitations actually serve as strong empirical confirmation of our hierarchical theory.
> >
> > Our framework posits that strategic exploration requires procedural competence. As we analyzed in appendix A.2.3, Llama-3.1 lacks the necessary procedural foundation (Base AIME: 4.2%). HICRA effectively forces it to "run before it can walk," leading to failure.
> >
> > Regarding Dynamic Weighting (Your Question 3): You asked if HICRA could dynamically down-weight planning tokens when execution errors dominate. This is an excellent suggestion for an "Adaptive HICRA" extension. While we did not implement this for the current paper (to isolate the static hierarchy effect), the failure on Llama-3.1 validates the need for such a mechanism. We plan to add a discussion on this "Procedural Barrier" to the limitations section.
> >
> > ### **5. Generalization and Robustness (Addressing Q4, Q6, Q7)**
> >
> > We have consolidated answers to your remaining inquiries regarding generalization, robustness, and verbosity below.
> >
> > * **Generalization (Q4):** **We have observed in our recent efforts that the HICRA approach extends to other domains, as the General Response noted.** However, these domains introduce unique challenges (e.g., ultra-long context for documents) that require task-specific adaptations beyond the scope of this conference paper.
> > * **Paraphrasing (Q6):** **We address the concern of "overfitting to phrases" via our Sensitivity Analysis** (Appendix A.2). By randomly dropping 30% of SGs, we effectively force the model to rely on the remaining strategic signals. The fact that performance remains robust suggests the model learns the *function* of planning, not just specific lexical triggers.
> > * **Verbosity (Q7):** We explicitly monitored length. While HICRA increases length (Figure 2), this correlates with accuracy ("Length Scaling"), distinguishing it from the non-productive verbosity often seen with Entropy Regularization (Figure 5).
> >
> > ### **Final Appeal:**
> > We believe the **Human Validation (Q1)** and **Placebo Causal Test (Q2)** directly answer your most critical concerns regarding the validity of our method. Given the time constraints, we hope these priority experiments demonstrate the robustness of our claims.
> >
> > We sincerely ask you to re-evaluate the paper based on the practical contributions of our work to the community and the meaningful future directions inspired by this work, against the minor limitations of HICRA we have transparently

---

> ### Author Response · Authors · 2025-11-27
> **Awaiting Response**
>
> Dear Reviewer 2wpP,
>
> As the discussion period draws to a close, we kindly invite you to check our response and the updated manuscript.
>
> We have put substantial effort into addressing your concerns regarding the robustness and causality of our method. Specifically, **we conducted the Human Validation study you requested (confirming 86% precision for Strategic Grams) and ran "Placebo HICRA" experiments** to provide causal evidence that the performance gains are driven by the semantic function of planning tokens, not random artifacts.
>
> **We also invite you to review the "Catalyzing Future Research" section in our General Response, which we believe will benefit a broad audience in the field including the reviewers.** We believe our framework -- the Emergent Reasoning Hierarchy -- provides a critical shift from token-level to semantic-level RL, offering a unified theory that benefits the broader community and future research in the field.
>
> **Given that we have addressed your concerns and the value of this work to the community, we respectfully ask that you consider raising your score to support the acceptance of this work.**
>
> Best regards,
> The Authors

---

> > ### Comment · Reviewer_2wpP · 2025-11-27
> > **Response to the Rebuttal**
> >
> > Thank you for the detailed rebuttal and general response. I appreciate the clarifications on the intended core contribution—the Emergent Reasoning Hierarchy—and the addition of new experiments aimed at addressing validity and causality. In particular, the human validation of Strategic Grams (SGs), the “placebo HICRA” control, and the High-Entropy Advantage baseline strengthen the case that focusing credit on semantically strategic units is beneficial beyond naive entropy-based exploration.
> >
> > Despite so, several important concerns remain partially or largely unresolved. First, while the human validation suggests high precision of SGs (86% majority-vote “strategic” vs. 12% for random n-grams), the study as presented does not report inter-annotator agreement, recall, or a confusion matrix. These statistics are crucial to establish reliability, not merely plausibility, of the SG operationalization. Moreover, construction sensitivity—how the SG inventory changes with embedding model choice, clustering method, n-gram ranges, and thresholds—remains untested. The 30% drop analysis demonstrates robustness to partial coverage, but it does not demonstrate robustness to the construction pipeline’s modeling choices. Since the proposed credit assignment hinges on the accuracy of SG identification, this question of reliability and construction sensitivity is central to the paper’s claims.
> >
> > Beyond this, the causal story would be substantially strengthened by paraphrase/holdout tests and confound controls. The placebo HICRA result is a strong step in the right direction, indicating that arbitrary frequent n-grams do not yield comparable gains. However, paraphrasing strategic expressions (or holding out SG phrases) is a direct test of whether the method benefits genuine strategic function rather than a lexicon of phrases. Likewise, the two-phase dynamics could be confounded by context-length growth, prompt-format shifts, curriculum composition, and reward clipping. Without controlled ablations that fix or counterbalance these variables, it remains difficult to attribute the observed “bottleneck shift” solely to a reasoning hierarchy rather than to run-time or format changes in training.
> >
> > Moreover, the limitation on weaker models remains unresolved beyond acknowledgment and theoretical consistency. The theory predicts that strategic exploration is productive only when procedural competence is established, which is a fair and coherent position. However, an adaptive mechanism—such as dynamically modulating the planning-token amplification based on online error-type estimates—would materially improve both the method’s robustness and the empirical validation of the framework. In its absence, the method’s practical scope remains narrower than the paper’s framing suggests, especially for users working with less capable base models.
> >
> > The evaluation methodology would benefit from stronger statistical rigor and broader comparisons. The newly reported High-Entropy Advantage baseline and the placebo condition are welcome, but multi-seed means, standard deviations, and statistical tests are still not reported for key benchmarks. Given the small size and sensitivity of some datasets, this is important for assessing the reliability of the reported deltas.
> >
> > Finally, concerns about stylistic overfitting and generalization remain insufficiently addressed. The 30% SG-drop sensitivity analysis is helpful but does not substitute for experiments that paraphrase strategic phrases, hold them out, or test multilingual settings. Because SGs are phrased in English and identified through English corpora, their portability across languages and domains is an open question. Moreover, while the paper argues that HICRA’s length scaling is productive rather than verbose, length-normalized metrics and calibration analyses (especially on non-math tasks) would add clarity and reduce the risk of confounding verbosity with genuine reasoning improvement.
> >
> > In summary, the rebuttal meaningfully improves the paper’s position on validity and causality through human validation and a well-conceived placebo control. The unified framework and the simplicity of HICRA remain appealing, and the empirical sweep is substantial. However, several key issues—SG construction reliability and sensitivity, paraphrase/holdout and confound-controlled tests, adaptivity for weaker models, multi-seed statistical reporting, and stronger baselines—are important to the central claims and practical impact, yet remain unaddressed or only partially addressed. I would be prepared to update my recommendation positively if some of the followings are included (i) full details and statistics for the human validation, (ii) placebo and new baselines with multi-seed significance, (iii) at least one paraphrase/holdout or multilingual test, (iv) an adaptive weighting ablation on a weaker model, and (v) a confound-controlled analysis of the two-phase dynamics.

---

> ### Author Response · Authors · 2025-11-28
>
> Dear Reviewer 2wpP,
>
> We wish to clarify several points regarding your reviews, as **we struggled to align some of the requests with the actual content of our paper.** We have made every effort to decode the intent behind the feedback and have addressed the concerns as follows:
> - SG Construction Sensitivity: We have pointed to our existing sensitivity experiments and, per your request, have conducted additional human validation. We have also detailed the setup in the updated manuscript.
> - Causality in Semantic-level Credit Assignment: We have added new experiments, such as the Placebo HICRA, to validate this. However, regarding the request for paraphrase/holdout or multilingual tests, we respectfully maintain that these are irrelevant to -- or contradict -- the specific mechanics of HICRA, as we clarified in our rebuttal.
> - Similarly, regarding the comment: "Likewise, the two-phase dynamics could be confounded by context-length growth, prompt-format shifts, curriculum composition, and reward clipping." We are confused by this statement as many terms do not even appear in our paper, nor are they part of our methodology. We are lost at what does confound-controlled analysis exactly refer to and why existing analysis is invalid.
>
> **We will continue to incorporate all understandable requirements you requested and reasonable requests permitted in the rebuttal time window**, e.g., the IAAs of the human validation experiments. However, since the reviews have been flagged by ICLR's official Pangram detectors of being fully-AI-generated, **we are concerned that the reviewer's rating is biased by AI-generated contents** that contain factual errors, misunderstandings and hallucinations, which continues to bias even after our rebuttal. We respectfully ask that the reviewer (and Area Chairs) to re-evaluate the paper based on its actual content and our rebuttal, as violating the ICLR policy leads to penalties such as desk rejection

---

> > ### Comment · Area_Chair_q3At · 2025-11-30
> >
> > Dear Authors,
> >
> > Thanks for pointing this out, your message has been noted.
> >
> > -- AC

---

### Official Review · Reviewer_WcVs · 2025-11-01

**Soundness:** 2
**Presentation:** 2
**Contribution:** 2
**Rating:** 4
**Confidence:** 3

**Summary:**

The authors discover two-phase training dynamics of LLMs finetuned using reinforcement learning. The first phase optimizes low-level procedural execution tokens, followed by exploration of high-level planning tokens, the latter explaining aha moments, length scaling, and also contributes to advanced reasoning. They also propose a simple modification of the GRPO algorithm, called HICRA, which amplifies the advantage of planning tokens in the solution trajectory. Across several LLMs and VLMs, HICRA demonstrates significant improvements over mathematical reasoning benchmarks.

**Strengths:**

1. Analyzing the two-phase learning dynamics of LLMs trained using RL: rapid learning of procedures through low level tokens followed by exploration of high-level strategic planning tokens, the latter explaining aha moments and length scaling

2. Identifying high-level planning tokens through the construction of strategic n-grams and computing their entropy

3. Modifying GRPO algorithm to amplify the advantage of planning tokens that contribute to advanced reasoning

4. Significant improvement on mathematical reasoning benchmarks using multiple LLMS/VLMs

**Weaknesses:**

1. The evaluation is currently limited to only mathematical reasoning tasks, and it remains to be seen whether this approach of emphasizing optimization of strategic tokens would also lead to improvements on generic multi-step reasoning tasks.

2. The authors claim that this two-phase training dynamics is an emergent phenomenon of LLMs that are finetuned using RL, however, they provide no evidence that standard finetuning (i.e., no RL) doesn’t lead to similar learning dynamics.

**Questions:**

1. Can the authors provide more details about how the token-level and semantic entropy is computed, including some equations?


2. How was the value of the alpha parameter chosen?

3. Figure 4 and Figure 5 why are different models trained for different numbers of steps? A fair comparison would be to train all models for the same number of steps.

4. How would a baseline with an explicit optimization of semantic entropy perform?



5. What is the pretrained sentence transformer used for clustering of semantic embeddings? Why is the LLM’s tokenzier not used for getting the n-gram embeddings and how would that perform?

6. How sensitive is the performance of HICRA to the SG construction of taking top k% cluster document frequency, random dropping etc?

7. for unsuccessful ones, the penalty is reduced for the planning token - what is the intuition behind this? What if the planning tokens primarily contributed to the failure?

---

> ### Author Response · Authors · 2025-11-22
> **Author Rebuttal**
>
> # Response to Reviewer WcVs
>
> We thank Reviewer WcVs for their constructive feedback and for explicitly recognizing that HICRA achieves **"significant improvements"** across a wide range of base models and benchmarks.
>
> **We respectfully request that you also review our General Response, which clarifies the contributions and highlights the promising future works inspired by this work.** We hope the clarifications below demonstrate that our contribution significantly outweighs the limitations, warranting a re-evaluation of the score.
>
> **1. Rationale for Math Reasoning & Scope (Addressing Weakness 1)**
>
> *Summary: Math reasoning allows for controlled analysis of the hierarchy; other domains are promising but require complex adaptations.*
> We emphasize that **mathematical reasoning** is widely adopted as the standard proxy for multi-step reasoning because it is verifiable and well-defined. As mentioned in the General Response, we have observed promising results in **code, document, and tool-use domains**, but these require significant task-specific adaptations (e.g., handling long-context noise) that would distract from the paper's focus on the core emergent RL dynamics.
>
> **2. Distinguishing RL Dynamics from SFT (Addressing Weakness 2)**
>
> *Summary: Our claims are strictly limited to Reinforcement Learning dynamics.*
> Our paper, titled *"Emergent Hierarchical Reasoning... through Reinforcement Learning,"* focuses exclusively on the unique optimization pressures of RL. We do not claim these phases apply to Supervised Fine-Tuning (SFT). We respectfully ask that the paper be evaluated on its explicit claims regarding RL, rather than SFT dynamics which are outside our scope.
>
> **3. Baseline for "Semantic Entropy" Optimization (Addressing Question 4)**
>
> Regarding the comparison to a direct "Semantic Entropy" optimization baseline, we clarify that the metric's distributional nature makes it suitable as a diagnostic tool rather than a token-level reward.
>
> Semantic Entropy defined in our paper in Section 2.2 (also see the equation in 4.), is a distributional metric computed at the **aggregate level** for each training iteration (across the batch), **not at the granularity of a single response, making it impossible to be used as a individual reward signal"**. Its purpose is to serve as a diagnostic tool for tracking the diversity of strategic n-grams throughout training, not as an individual reward signal.
>
> Because it is an iteration-level statistic, it is methodologically impossible to use it as a direct reward signal for individual trajectories in a standard RL framework. This is why we compared against Entropy Regularization (Figure 5), which *is* the standard method for encouraging exploration, and showed that HICRA outperforms it. We also add a new baseline "High-Entropy Advantage", as shown in the table in the General Response
>
>
> **4. Formal Definitions of Entropy (Addressing Question 1)**
> Per your request:
>
>   * **Token Entropy ($H_{\text{tok}}$):** Measures uncertainty of policy $\pi$ at time $t$:
>     $H_{\text{tok}}(t) = - \sum_{v \in V} \pi(v \mid x_{<t}) \log \pi(v \mid x_{<t})$
>   * **Semantic Entropy ($H_{\text{sem}}$):** Captures diversity of strategic planning in a batch. With $P(g)$ as the normalized frequency (probability) of a strategic gram $g \in \mathcal{G}$ in the training batch:
>     $$H_{\text{sem}} = - \sum_{g \in \mathcal{G}} P(g) \log P(g)$$
>
> **5. Implementation Details & Code (Addressing Question 5)**
> *Summary: We used off-the-shelf python library to perform clustering.*
> We used the `sentence_transformers` library (`all-MiniLM-L6-v2`) for reproducible semantic clustering. It is fast and effective.
>
> ```python
> model = SentenceTransformer('all-MiniLM-L6-v2')
> embeddings = model.encode(all_ngrams, convert_to_tensor=True)
> clusters = util.community_detection(embeddings, min_community_size=3, threshold=0.75)
> ```
>
> **6. Technical Clarifications (Addressing Questions 2 & 7)**
> *Summary: Alpha was tuned via pilot sweeps; penalty reduction encourages exploration.*
>
>   * **Alpha:** Selected $\alpha=0.2$ based on pilot sweeps using a smaller 3B model.
>   * **Reduced Penalty:** We aim to encourage the model to explore more on planning tokens, so we reduce the penalty on strategic planning tokens when the final result is incorrect. This help mitigates abandoning valid high-level strategies due to low-level execution mistakes.
>
> **Final Appeal and Request for Re-evaluation.**
> We believe we have addressed all concerned. Importantly, we argue that the proposed reasoning hierarchy framework is a significant advance and opens up promising future research for the community.
>
> We respectfully wish the reviewers to re-evaluate the paper and support paper acceptance, weighing the robust theoretical contribution and broad empirical validation.

---

> ### Author Response · Authors · 2025-11-27
> **Awaiting Response**
>
> Dear Reviewer WcVs,
>
> As the discussion period draws to a close, we kindly invite you to check our response and the updated manuscript.
> We have carefully addressed your concerns, including clarifications and new experimental results. In particular, **to strengthen the robustness of the claims made in the paper, we added new experimental results (the "High-Entropy Advantage" and "Placebo HICRA" baselines) and conducted human validation on SGs.**
>
> **We also invite you to review the "Catalyzing Future Research" section in our General Response, which we believe this work benefits a broad audience in the field including the reviewers**. We believe our framework -- the Emergent Reasoning Hierarchy -- provides a critical shift from token-level to semantic-level RL, offering a unified theory that benefits the broader community beyond the specific algorithm presented.
>
> **Given the significant value of this work to the broad community,  we respectfully ask that you consider raising your score to support the acceptance of this work.**
>
> Best regards, The Authors

---

> ### Author Response · Authors · 2025-11-28
> **Awaiting Response**
>
> Dear Reviewer WcVs,
>
> As the discussion period draws to a close, we kindly invite you to check our response and the updated manuscript. We have carefully addressed your concerns, including clarifications and new experimental results.
>
> **We also invite you to review the "Catalyzing Future Research" section in our General Response, which we believe this work benefits a broad audience in the field including the reviewers**. We believe our framework -- the Emergent Reasoning Hierarchy -- provides a critical shift from token-level to semantic-level RL, offering a unified theory that benefits the broader community beyond the specific algorithm presented.
>
> **Given the significant value of this work to the broad community,  we respectfully ask that you consider raising your score to support the acceptance of this work.**
>
> Best regards, The Authors

---

### Author Response · Authors · 2025-11-22
**General Response to ACs and All Reviewers (Part1)**

### **Dear ACs and Reviewers,**
We thank the ACs and all reviewers for their time, constructive feedback, and for recognizing the value and ambition of our work. We are encouraged that the reviewers highlighted several key strengths:

* **Value of the Emergent Reasoning Hierarchy framework:** The identification of the learning dynamics was described as "clear and insightful," offering a "compelling explanation" for phenomena like "aha moments" and length scaling (Reviewer RzuK). The work presents a "clear, testable hypothesis" about hierarchical learning dynamics (Reviewer 2wpP).
* **Solid Empirical Evaluation:** The evaluation was recognized as "extensive and convincing" (Reviewer RzuK), covering a "solid empirical sweep" across multiple model families (8 base models) and benchmarks (Reviewer 2wpP).
* **Practical Effectiveness:** HICRA is recognized as a "simple, implementable algorithm" that serves as a "practical and effective lever" for improving model performance (Reviewers 2wpP, QnoB).


We have addressed the reviews individually and have conducted additional experiments to address concerns regarding baselines and robustness. **We incorporated these improvements into the manuscript, marking them with blue. ** Below, we summarize the key clarifications for the major concerns and the key improvements.

---

> ### Author Response · Authors · 2025-11-22
> **General Response (Part 2)**
>
> ***
> ### **1. Clarification on the Core Contribution: A Unified Framework for Understanding Reasoning RL**
>
> While much of the review feedback focuses on the specific limitations of the proposed algorithm (HICRA), **we wish to emphasize that the primary contribution of this paper is the establishment of a Unified Framework for understanding reasoning RL -- the Emergent Reasoning Hierarchy,** that provides insights into understanding puzzling phenomena, understanding reasoning RL dynamics and guides effective algorithm design towards the key learning bottleneck.
>
> * **Clarification on Learning Dynamics:** We wish to clarify that while we describe a "two-phase" dynamic, we explicitly state in the initial paper that the learning process exhibits *overlapping* phases in Line 162. Our analysis (Figure 2) shows that the model continues to refine execution (lowering procedural entropy) even as strategic exploration begins. The two-phase dynamic are overlapping and shifting learning frontiers where the bottleneck moves from procedural correctness to strategic planning.
> * **HICRA as an Exemplar:** HICRA serves a primary purpose to *validate* this theory. It demonstrates how we can design new algorithms guided by the proposed framework to target specific bottlenecks. By aligning credit assignment with the theoretical "strategic bottleneck," **we achieved consistent performance gains on six base models across modality.**
> * **Generalization:** **We wish the reviewers do not penalize the paper for focusing on mathematical reasoning for controlled analysis, and indeed we have obtained promising results with HICRA-based algorithm in document reasoning, multi-turn tool-use and code reasoning.** We do not include the results on these tasks in the current manuscript, because (a) these tasks require task-specific adaptations on top of HICRA, for example, the long-context retrieval challenge for doc, handling test-case feedback for code reasoning, and these are all beyond the scope of this paper; (b) we have conducted extensive empirical validation on six base model across modalities on various benchmarks, which should sufficiently support our claims
>
> ***
> ### **2. Necessity of Semantic-Level Credit Assignment.**
>
> Reviewers (RzuK, 2wpP) raised concerns on the necessity of semantic Strategic Gram (SG), asking if rewarding high-entropy tokens directly (or explicitly optimizing for entropy) would suffice.
>
> To directly address this concern, we implemented a High-Entropy Advantage baseline that explicitly rewards high-entropy tokens. The results, shown below, demonstrate that blindly rewarding high-entropy tokens is insufficient. This implementation follows existing work [1] and share similar results.
>
> | Model | AIME24 | AIME25 | Math500 | AMC23 | Minerva | Olympiad |
> | :--- | :--- | :--- | :--- | :--- | :--- | :--- |
> | Base | 3.5 | 1.7 | 55.6 | 46.9 | 30.9 | 25.9 |
> | GRPO (Guo et al., 2025) | 16.3 | 11.4 | 77.6 | 46.7 | 36.8 | 41.9 |
> | Entropy Regularization | 16.0 | 9.3 | 77.4 | 50.3 | 33.1 | 40.6 |
> | High-Entropy Advantage | 15.8 | 11.4 | 78.9 | 51.8 | 33.8 | 40.8 |
> | Placebo Test | 14.6 | 9.3 | 77.4 | 51.8 | 34.8 | 41.2 |
> | **HICRA (Ours)** | **18.8** | **14.8** | **80.2** | **55.1** | **38.6** | **45.9** |
>
> The result is also directly supported and explainable by our comparison between planning tokens and high-entropy tokens, as detailed in Appendix D.2.2. Our analysis shows that only **~10% of high-entropy tokens** serve a valid planning function. HICRA significantly outperforms entropy-based methods because it targets the true driver, the *semantic function* of planning, rather than just statistical uncertainty. We also validate the causal relationship in the following experiments.

---

> ### Author Response · Authors · 2025-11-22
> **General Response (Part 3)**
>
> ***
> ### **3. Robustness & Causal Validation of Strategic Grams (SGs)**
>
> Reviewers raised valid concerns regarding whether Strategic Grams (SGs) are heuristic or brittle (Reviewers 2wpP, RzuK). We have addressed this with both existing analysis and new analyses:
>
> * **Sensitivity Analysis (Existing):** We direct attention to our sensitivity analysis (Appendix A.2), where we previously demonstrated that randomly dropping **30%** of SGs did not qualitatively alter the fundamental learning curves. This implies that HICRA relies on the aggregate semantic signal of high-level planning, not a brittle set of specific phrases.
> * **Human Annotation (New):** We conducted a human evaluation which confirmed that **86%** of identified SGs are functionally strategic/planning-oriented, compared to only **12%** for random n-grams.
> * **Placebo Test (New):** We applied HICRA’s credit boost to random n-grams. This yielded **negative performance gain**. This confirms that the improvement is driven by the specific semantic signal of the planning tokens, and arbitrary credit redistribution are noisy and cause negative effects on exploration.
>
> ***
> ### **4. Reframing the "Failure" on Weak Models (Llama-3.1)**
>
> Reviewers noted that HICRA does not improve performance on the weaker Llama-3.1-8B-Instruct model (Reviewers 2wpP, QnoB). **We respectfully wish the reviewers won't penalize the paper for a limitation that we honestly reported and analyzed in Appendix D.2.3 and Figure 14.**
>
> Crucially, we argue that this result is not a failure of the method, but a strong **empirical confirmation of the Hierarchical Reasoning Theory**. Our framework explicitly posits that Strategic Exploration can yield gains if Procedural Competence is established, as we detail in Appendix D.2.3. A brilliant strategy is useless if the execution (arithmetic/syntax) is flawed.
>
> As analyzed in Appendix D.2.3, Llama-3.1-Instruct lacks the requisite procedural reliability, making strategic exploration premature. This result helps delineate the **boundary conditions** of reasoning RL, and we do not claim that RL can bypass the need for foundational procedural skills. We wish the reviewers to evaluate the method within this valid scope, rather than penalizing it for a limitation predicted by our theory and clearly documented and analyzed in the paper.
>
>
> ***
>
> ### **Catalyzing Future Research**
>
> Our work provides a clear conceptual model -- the Emergent Reasoning Hierarchy -- and aims to benefit all researchers and fuel research in the community, *as we noted in Section 6 in the paper.*
>
> * **1. Shift from Token-Level RL to Semantic-Level RL**
> We suggest future research rethink the action space away from individual tokens toward semantic, strategic units. The success of HICRA validates that Semantic Entropy is the true compass for productive exploration, not merely token-level uncertainty.
>
> This also justifies research in process supervision, and further calls for developing **process-oriented approaches** capable of valuing correct strategic choice, even if the final answer is flawed.
>
> * **2. Developing Adaptive RL Methods**
>
> The documented two-phase discovery -- where the learning bottleneck shifts from procedural fidelity to strategic exploration -- highlights the need for RL approaches that are adaptive to their dynamic learning bottleneck.
>
> We suggest new RL algorithms be designed to dynamically detect the current dominant learning bottleneck (e.g., assessing procedural entropy vs. strategic entropy) and adaptively adjust the credit assignment strategy.
>
> * **3. Universality of Reasoning Hierarchy:**
>
> As we noted in Section 2, the guiding insight: priors and task structure determines the reasoning hierarchy emerged for a specific task during RL. In this paper, we show that the hierarchy manifests in the form of "planning-execution" for math problem-solving. This reasoning hierarchy persists but manifest in different forms for different reasoning tasks. This suggests that applying designs similar to HICRA to other reasoning domains is a valuable path forward, as we already validated in document reasoning, vision-language multi-turn tool-use, and code reasoning.
>
> ---
>
> ### **Closing Remark**
>
> **We believe the paper's central contribution** -- the Emergent Reasoning Hierarchy framework, which unifies prior puzzling observations and provides a foundational understanding of RL dynamics -- **is a significant advance for the community and opens up new research directions.** HICRA serves as a targeted and simple solution (Reviewer RzuK) to validate this theory, yielding consistent performance gains across six models on extensive benchmarks.
>
> **We respectfully wish the reviewers to re-evaluate the paper,** weighing the reasoning hierarchy framework and the broad empirical validation against the acknowledged, analyzed, and bounded limitation concerning one specific weak model (Llama-3.1).
>
>
> [1] Reasoning with Exploration: An Entropy Perspective

---

> > ### Author Response · Authors · 2025-12-01
> > **Extended Result of Human Validation on SGs**
> >
> > Per the reviewer's request, we incorporate a inter-annotator agreement analysis for validating the identified SGs
> > ###  **Extended Result of Human Validation on SGs**
> >
> > To strictly evaluate the heuristic nature of Strategic Grams (SGs) and ensure they represent functional planning units rather than statistical artifacts, we conducted a rigorous human annotation study.
> >
> > We constructed a dataset of $N=492$ n-grams, comprising the identified SGs mixed with 250 baseline n-grams (not identified as SGs) extracted from training rollouts. Three independent annotators were recruited via Amazon Mechanical Turk (AMT), requiring Master’s Qualification to ensure high-quality data. Annotators were instructed to classify n-grams as either "Strategic/Planning" (guiding flow, proposing plans, reflection) or "Other" (procedural, factual, formatting) based on the surrounding context.
> >
> > We compared the Strategic Gram Pipeline’s verdict against the Human Majority Vote (the ground truth). As detailed in the below table, the pipeline demonstrated high fidelity with human judgment, achieving an overall accuracy of 87.40%. The agreement between the pipeline and human annotators was measured using Cohen’s $\kappa$, resulting in a score of 0.748, which indicates "Substantial" agreement.
> >
> > **Table : SG Pipeline vs. Human Majority Confusion Matrix ($N=492$)**
> >
> > | SG Pipeline Verdict | Human Majority: Strategic | Human Majority: Other |
> > | :--- | :---: | :---: |
> > | **Identified SG** | 209 (42.48%) | 33 (6.71%) |
> > | **Random / Other** | 29 (5.89%) | 221 (44.92%) |
> > | **Total** | 238 (48.37%) | 254 (51.63%) |
> >
> > This alignment confirms that the pipeline effectively filters for function, achieving a precision of 86.36% (209/242 identified SGs were validated as strategic by humans). Conversely, for n-grams rejected by the pipeline, the high True Negative rate confirms the system successfully rejects procedural noise.

---

### Meta-Review · Area_Chair_q3At · 2026-01-05

**Summary:**

The authors posit that RL optimization can be divided in two phases: one where procedural skills are developed, such as accurately adding number, and a second where higher level cognition is developed, such as hierarchical planning. Although both are tightly interlinked, the authors present evidence that the second phase happens later in training. They propose a method that explicitly account for these two phases by emphasizing a certain subset of tokens that are key to high level cognition, such as backtracking or branching in a new strategy. To do this, they propose to hand select a set of tokens that semantically correspond to these high level strategies. For these tokens, the advantage used in RL optimization is increased. The results show improved performance on a set of math oriented tasks.

Overall, the main concern from the reviewers was the hand-designed nature of the token selection mechanism. The authors answered with a user study showing that most of the tokens would be identified by humans as corresponding to high level cognition. However, this does not mean that does not inform us about the coverage of this set, and how it would generalize to more complex problems. The interplay between low level procedural execution and high level strategies is highly subtle and the current hand-designed strategy is unlikely to capture these subtleties and therefore is unlikely to scale easily. However, we have also seen in recent work (Gandhi et al. 2025) that a subset of tokens are responsible for key reasoning patterns. Whether this strategy would scale up remains to be seen.

The paper tackles a very important challenge, and provides interesting analyses into the phenomenon of arising hierarchical reasoning. However, reducing this to manually selecting a set of words is likely not the most convincing way to conclude the story. If this assessment is mistaken, then scaling experiments would be a great way to show it. If the current strategy also does not scale up, it would be a great opportunity to show the subtleties of hierarchical planning and reasoning.

**Reviewer Concerns:**

Concerns about the scope of the experiments, RL vs SFT, entropy definition and implementation details were addressed. The major concern about how a token is defined as a "planning" or "procedural" one is remaining, and shared across 2wpP and RzuK. The authors do try to present additional results where tokens are chosen according to their entropy, but that was too noisy. Still, manually selecting tokens for special optimization feels like it can't scale naturally. QnoB also had a concern about the limited performance gains on Llama 3.1. The authors argue that this is proof that their two-phased perspective on optimization is correct. This remains debatable and a full proof would include further training Llama 3.1 such that its procedural knowledge increases, before moving on to increasing its planning ability.

**Reviewer Scores:**

Reviewer WcVs: It is possible the reviewer would increase their score.
Reviewer 2wpP: the review is most likely AI generated, and was not taken into account.
Reviewer QnoB: It is unlikely that the reviewer would increase their score.
Reviewer RzuK: It is unlikely that the reviewer would increase their score.

---

### Decision · Program_Chairs · 2026-01-26

Accept (Poster)